# Covalent organic frameworks for direct photosynthesis of hydrogen peroxide from water, air and sunlight

Fuyang Liu [1,2,3,7], Peng Zhou [4,7], Yanghui Hou[1,2,3], Hao Tan[5], Yin Liang[5], Jialiang Liang[6], Qing Zhang [5], Shaojun Guo [5], Meiping Tong [1,2,3] ✉ & Jinren Ni [1,2,3]

Solar-driven photosynthesis is a sustainable process for the production of hydrogen peroxide, the efficiency of which is plagued by side reactions. Metal-free covalent organic frameworks (COFs) that can form suitable intermediates and inhibit side reactions show great promise to photo-synthesize $H_2O_2$. However, the insufficient formation and separation/transfer of photo-generated charges in such materials restricts the efficiency of $H_2O_2$ production. Herein, we provide a strategy for the design of donor-acceptor COFs to greatly boost $H_2O_2$ photosynthesis. We demonstrate that the optimal intra-molecular polarity of COFs, achieved by using suitable amounts of phenyl groups as electron donors, can maximize the free charge generation, which leads to high $H_2O_2$ yield rates (605 μmol $g^{-1}$ $h^{-1}$) from water, oxygen and visible light without sacrificial agents. Combining in-situ characterization with computational calculations, we describe how the triazine N-sites with optimal N *2p* states play a crucial role in $H_2O$ activation and selective oxidation into $H_2O_2$. We further experimentally demonstrate that $H_2O_2$ can be efficiently produced in tap, river or sea water with natural sunlight and air for water decontamination.

As a versatile reagent, hydrogen peroxide ($H_2O_2$) is widely used in the fields of chemical synthesis, energy storage and water treatment[1,2]. At present, oxidation of anthraquinone, electrochemical synthesis, and noble metal catalysis are the common methods used to produce $H_2O_2$[2,3]. However, these methods require high energy input and cause environmental pollution by the release of toxic by-products[4,5]. Due to the use of naturally abundant water and oxygen in air as raw materials and solar light as energy input, $H_2O_2$ photosynthesis, especially without using sacrificial agents, is regarded as one of the green and sustainable methods, which yet is plagued by the severe side reactions such as the decomposition of produced $H_2O_2$ into oxygen and water

due to the metastable feature of $H_2O_2$ during the synthesis process[6–12]. As one type of novel metal-free crystalline polymers that can form suitable intermediates and inhibit side reactions for $H_2O_2$ generation, covalent organic frameworks (COFs) recently show great promise in the field of $H_2O_2$ photosynthesis[13,14]. The insufficient formation or separation of excitons (bound state of electron–hole pairs) for the free charge generation in COFs yet seriously constrains the high-efficiency of $H_2O_2$ photocatalytic generation[15,16]. Some interfacial modulating strategies such as constructing heterojunction[17] and incorporating single atoms[18] are beneficial for enhancing the photoexcitation and charge separation/transfer of COFs photocatalysts, which yet require

[1]College of Environmental Sciences and Engineering, Peking University, Beijing 100871, PR China. [2]The Key Laboratory of Water and Sediment Sciences (Ministry of Education), Peking University, Beijing 100871, PR China. [3]State Environmental Protection Key Laboratory of All Material Fluxes in River Ecosystems, Peking University, Beijing 100871, PR China. [4]School of Environment and Energy, Peking University Shenzhen Graduate School, Shenzhen, Guangdong 518055, PR China. [5]School of Materials Science and Engineering, Peking University, Beijing 100871, PR China. [6]College of Environment and Ecology, Chongqing University, Chongqing 400045, PR China. [7]These authors contributed equally: Fuyang Liu, Peng Zhou. ✉e-mail: tongmeiping@pku.edu.cn

multistep, complex and time/energy-consuming synthesis process. Hence, developing a facile and cost-effective strategy to improve the generation and separation of photogenerated charges without introducing severe side reactions is in great demand for $H_2O_2$ photosynthesis, yet remains a great challenge.

Herein, we provide a facile and economical strategy for the design of metal-free donor-acceptor (D-A)-type COFs with optimal intramolecular polarity by introducing suitable amounts of phenyl groups as electron donors for excitonic regulation to boost the direct photocatalytic $H_2O_2$ production from water, air and sunlight without using sacrificial agent. By using triazine-cored triamine with different amounts of phenyl groups ($n = 0, 1, 2$) as the precursors, a class of D-A COFs with different intramolecular polarity were successfully synthesized. We find that weak intramolecular polarity in D-A COFs constrains excitons dissociation, yet too strong intramolecular polarity inhibits excitons formation via weakened π-conjugated effect as well as decreases the photo-stability of COFs. The well-designed COFs with the optimal intramolecular polarity (named as COF-N32) can facilitate excitons formation and dissociation, leading to the high and stable $H_2O_2$ yield (605 μmol g$^{-1}$ h$^{-1}$) with solar-to-chemical efficiency of 0.31% in water without additional sacrificial reagent. COF-N32 can also efficiently produce $H_2O_2$ in real water samples including tap water, river water and sea water even with natural solar light irradiation. Moreover, COF-N32 can be assembled into practical devices for facile consecutive uses with high photocatalytic stability under natural solar irradiation. In addition, we demonstrate that the produced $H_2O_2$ aqueous solution (without further separation) can be directly employed in water decontamination, indicating the potential application feasibility of COF-N32. Via in-situ Fourier transform infrared (FTIR) characterization and density functional theory (DFT) calculation, we reveal that the suitable N $2p$ states and C $2p$ states in COF-N32 with optimal intramolecular polarity effectively reduce the energy barrier for $H_2O$ activation and oxygen reduction, respectively, contributing to the high efficiency.

## Results and discussion
### Characterization of synthesized COFs
The COFs photocatalysts were synthesized by a solvothermal method. Figure 1a illustrates the theoretical chemical structures of three triazine-based COFs, which is also confirmed by the corresponding solid state $^{13}C$ nuclear magnetic resonance (NMR) spectra (Fig. 1b). Clear peaks at ~180 ppm, ~170 ppm, ~145 ppm and ~115 ppm in the $^{13}C$ NMR spectra of three COFs can be attributed to carbonyl carbon, triazine carbon, C-NH (amine linkage) carbon and olefin carbon, respectively[19]. The stretching vibration bands of C=C bond at ~1573 cm$^{-1}$ and C-N bond at 1255 cm$^{-1}$ in the FTIR spectra of three COFs (Fig. 1c) suggest the occurrence of Schiff base reaction and enol-to-keto tautomerism during the fabrication process of three COFs[19,20]. The formation of carbonyl group and amine linkage in three COFs is further confirmed by X- photoelectron spectroscopy (XPS) analysis (Fig. S1). Based on the simulation and Pawley refinement, the obvious peaks at $2\theta = 9.6°$, 5.5° and 3.9° corresponding to (100) planes and $2\theta = ~27°$ corresponding to (002) planes in the X-ray diffraction (XRD) pattern (Figure S2) suggest that three COFs contain crystalline structure[20-22]. Scanning electron microscopy (SEM) and transmission electron microscopy (TEM) images show that the three COFs are tiny granular particles with diameter of 2–3 μm, which are assembled by numerous nanorods (Figs. S3 and S4). The above results confirm the successful fabrication of three COFs.

To explore whether the incorporated phenyl group on triazine rings would affect the formation and separation of excitons in COFs, the intramolecular polarity of electronic structure in three COFs was determined. The theoretical calculation of charge distribution indicates that in the hexatomic rings (one type of octupolar subunit), carbonyl group with *meta* position acts as electron acceptor, while

olefin group serves as donor (Figs. 1d and S5)[23-25]. For another type of octupolar subunit, due to the well-delocalizing π-electron over the three aromatic carbon atoms, 1,3,5-triazine ring with electron deficiency in all three COFs can also serve as electron acceptor center[26,27]. While enamine group in three COFs as well as phenyl group in COF-N32 and COF-N33 can act as electron-donating groups[28,29]. Clearly, all three COFs contain octupolar conjugated structure with two subunits (Fig. 1e), which is expected to facilitate the efficient charge separation especially in each subunit coupling with appropriate intramolecular polarity[27]. Accordingly, stable radicals with strong signal intensities are observed in the solid-state electron spin resonance (ESR) spectra of three COFs under dark condition at room temperature (Fig. S6). The paramagnetic absorption signal intensities of three COFs follow the order of COF-N31 > COF-N32 > COF-N33 (Fig. S6), indicating that different amounts of unpaired electrons exist in three D-A COFs under dark condition[30]. With increasing amount of phenyl groups as electron donors from 0 (COF-N31) to 1 (COF-N32) and further to 2 (COF-N33), the intramolecular polarization of composition fragments in three COFs decrease from 0.072 e Å$^{-1}$ for COF-N31 to 0.032 e Å$^{-1}$ for COF-N32 and further to 0.020 e Å$^{-1}$ for COF-N33 (Fig. S7 and Table S1). The direction of intramolecular polarity in these COFs is from enamine and/or benzene groups to triazine groups (Fig. S7). Furthermore, the molecular polarity index (MPI) also follows the same order of COF-N31 (0.53 eV) > COF-N32 (0.50 eV) > COF-N33 (0.48 eV) (Table S2).

The solvatochromic behaviors of COFs were employed to further determine their polar properties. Obvious shift of emission peaks (~48 nm) is observed in fluorescence spectra of COF-N31 dispersed in water (as a polar solvent) relative to those in methanol, dichloromethane and ethyl acetate (with weaker polarity than water), indicating the strong local dipolar nature of COF-N31 with weak electron donor in π-conjugated system[31]. Similar observation has also been previously reported for COFs with high polarity[27,32]. In contrast, for both COF-N32 and COF-N33, relatively smaller positive shifts of emission peaks (<25 nm) are observed in water relative to other organic solvents (Fig. S8). The weak solvatochromic behaviors of both COF-N32 and COF-N33 can be attributed to the sufficient electron donor with phenyl or diphenyl group in the octupolar π-conjugated framework. The COFs with higher polarity usually exhibit higher affinity to polar $H_2O$[33]. Note the intramolecular polarity follows the order of COF-N31 > COF-N32 > COF-N33. Accordingly, the contact angle of three COFs follows the order of COF-N31 (121°) < COF-N32 (139°) < COF-N33 (145°) (Fig. S9). Meanwhile, the unit water adsorption capacity is also consistent with the order of COF-N31 > COF-N32 > COF-N33 (Fig. S10).

In general, the excessively strong intramolecular polarity in COFs would restrain the π conjugated effect in $sp^2$-hybridized orbit structure of COFs, leading to the inhibition of electron excitation[34,35]. The O atom with high electronegativity in COF-N31 with strong polarity can attract electrons surrounding amine N atoms, the delocalization on triazine rings in the lowest unoccupied molecular orbital (LUMO) of COF-N31 thus is limited across three atoms (N and C atoms) in each direction (Fig. S11a). In contrast, due to the less polarity relative to COF-N31, LUMO of COF-N32 and COF-N33 can be well delocalized across nine atoms surrounding the triazine rings (Figs. S11b and S11c), indicating the enhanced π conjugation in COF-N32 and COF-N33. This would facilitate the excitation of electrons in these two COFs under light irradiation. The less intramolecular polarity of COFs has shown to result in the overlap of the highest occupied molecular orbital (HOMO) and LUMO[36,37]. Similarly, for COF-N32 and COF-N33 with less polarity, the overlap of HOMO and LUMO especially on the C atoms in keto structure and diphenyl group is also observed (Fig. S11). $S_m$ index (parameter denoting the hole-electron recombination degree[38]) of three COFs in the excited state based on DFT calculation is found to follow the order of COF-N31 (0.33) < COF-N32 (0.36) < COF-N33 (0.38) (Table S2). The observation indicates that hole-electron in COF-N32 and COF-N33 is

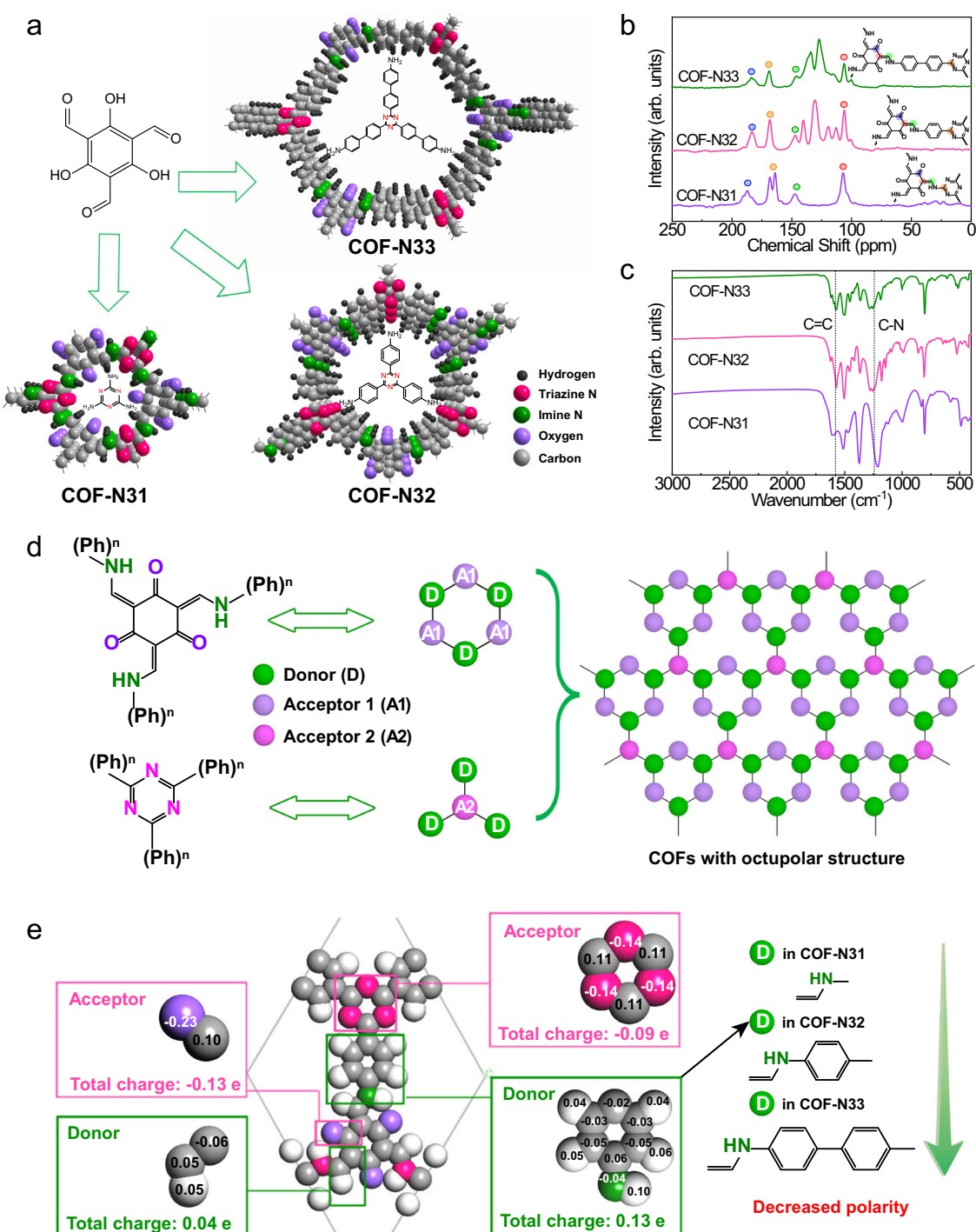

**Fig. 1 | Chemical structure. a** Schematic illustration of synthesis process of COF-N31, COF-N32 and COF-N33. **b** [13]C nuclear magnetic resonance (NMR) spectra and **c** Fourier transform infrared spectroscopy (FTIR) of COF-N31, COF-N32 and COF-N33. **d** The schematic illustration of the octupolar structure in COF-N31, COF-N32 and COF-N33. **e** DFT calculation on charge distribution of COF-N32 structure.

relatively readily to be recombined than COF-N31. Overall, the strong intramolecular polarity in COF-N31 theoretically inhibits the π conjugation, the subsequent excitation of electrons, as well as the recombination of charges. In contrast, the weak polarity in COF-N33 yet facilitates the π-conjugated effect, the excitation of electrons, and charge recombination. COF-N32 with moderate polarity among three COFs is expected to display the best photocatalytic property.

The formation, transfer and separation of photo-induced excitons in three COFs were investigated. Tauc plot based on UV-vis diffused reflectance spectra (DRS) reveals that all three COFs exhibit n−π*

transition in N atoms[39], while COF-N33 with more benzene units also contains more obvious signals of π−π* transition[40]. The band gaps of COF-N31 (2.72 eV) > COF-N32 (2.42 eV) > COF-N33 (2.40 eV) (Fig. 2a) are also consistent with the results of calculated ones (Fig. S12). The observation indicates that COF-N32 and COF-N33 exhibit higher light absorption efficiency relative to COF-N31, which can be attributed to the efficient π-conjugated effect with relatively weaker intramolecular polarity of these two COFs than that of COF-N31[27]. The results of electrical impedance spectra (EIS) indicate that COF-N32 and COF-N33 have relatively smaller charge transfer resistance than COF-N31

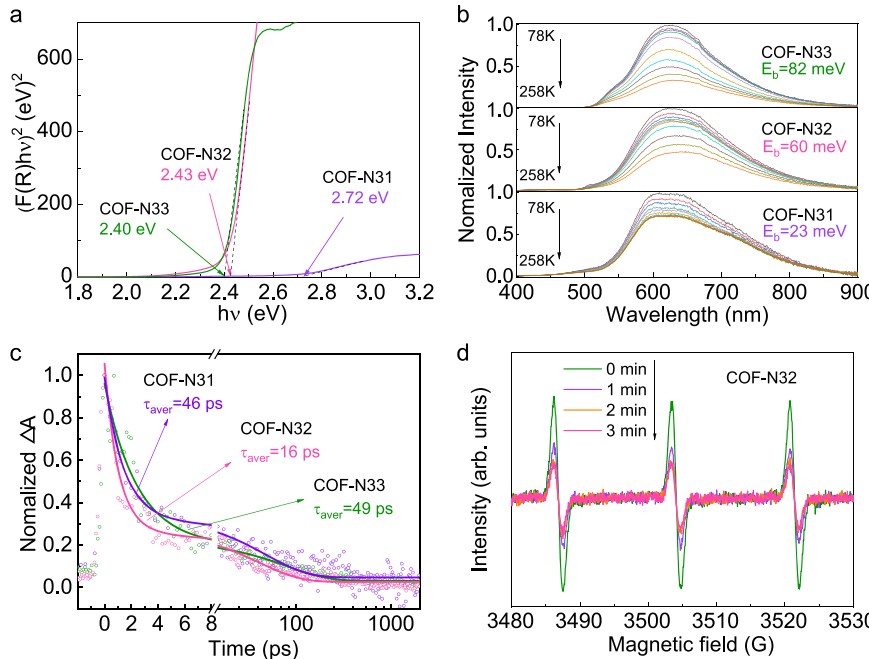

**Fig. 2 | Polarity property. a** Tauc plot of three COFs; **b** Temperature-dependent PL spectra of three COFs excited at 365 nm for the determination of binding energy ($E_b$). **c** Photoinduced absorption decay dynamics of three COFs with the excitation of 360 nm pump pulse (P = 4.3 mJ cm$^{-2}$ per pulse); **d** The signal intensity of 2,2,6,6-tetramethyl-1-piperidinyloxy (TEMPO) for charge detection in COF-N32 under light irradiation.

(Fig. S13), contributing to the efficient interfacial electron transfer speed of these two COFs. The observation agrees with the efficient π-conjugation through electronic push-pull effect in COF-N32 and COF-N33 due to their relatively weaker intramolecular polarity discussed above. Both the exciton binding energy ($E_b$) determined by temperature-dependent photoluminescence (PL) spectra and the emission peaks intensities in steady-state PL spectra of three COFs follow the order of COF-N33 > COF-N32 > COF-N31 (Figs. 2b, S14 and S15), indicating that the dissociation of formed excitons in three COFs is in the order of COF-N31 > COF-N32 > COF-N33.

The transient absorption (TA) was performed to analyze charge carrier dynamics in three COFs. The broad and positive absorbance changes (ΔA) observed around 500–700 nm can be attributed to the photoinduced absorption of photoexcited electrons in the conduction band (CB) of three COFs (Fig. S16)[41,42]. Among three COFs, photo-excited electrons in COF-N32 exhibits smaller average lifetime (16 ps) relative to those in COF-N31 (46 ps) and COF-N33 (49 ps) (Fig. 2c), which is consistent with the average lifetimes obtained in PL decay curves in the nanosecond domain (0.31 ns for COF-N32 relative to 0.75 ns for COF-N31 and 1.34 ns for COF-N33) (Fig. S17). The observations suggest the more prominent non-radiative rate in COF-N32 than COF-N31 and COF-N33[27,43]. ESR analysis (Fig. 2d) shows that under visible light irradiation, COF-N32 can generate more photo-induced free charges (via the excitons formation and dissociation) than COF-N31 (Fig. S18a) and COF-N33 (Fig. S18b). The generation of photo-induced free charges in COF-N32 is confirmed by detecting the reduction product tetramethylpiperidine (TEMP, m/z 142.15826, ESI+) via reaction of electrons and 2,2,6,6-tetramethyl-1-piperidinyloxy (TEMPO) (Fig. S19). The above results show that among all three COFs, COF-N32 with moderate intramolecular polarity in the octupolar conjugated structure can generate the greatest amount of charges (i.e. electrons and holes) under visible light irradiation (Eq. S4). The energy band positions of three COFs derived from XPS valence band spectra combined with Tauc plot indicate the thermodynamic feasibility of oxygen reduction reaction (ORR) to photosynthesize $H_2O_2$ (−0.33 V vs. NHE) by three COFs (−1.09 V vs. NHE for COF-N31, −0.55 V vs. NHE for COF-N32 and −0.48 V vs. NHE for COF-N33) (Figs. S20 and S21)[8], while

the two-electron water oxidation reaction (WOR) directly to $H_2O_2$ (1.77 V vs. NHE) is thermodynamically feasible by COF-N32 (1.88 V vs. NHE) and COF-N33 (1.92 V vs. NHE) but not by COF-N31 (1.63 V vs. NHE) (further discussion is provided below).

## Photocatalytic $H_2O_2$ production by COFs

The photosynthesis of $H_2O_2$ by COFs was evaluated in pure water without using any sacrificial agent under visible light irradiation (λ > 420 nm, 100 mW cm$^{-2}$). During 12 h reaction duration, COF-N32 with moderate intramolecular polarity in the octupolar conjugated structure exhibits significantly improved performance for $H_2O_2$ photosynthesis relative to COF-N31 (with strong intramolecular polarity) and COF-N33 (with weak intramolecular polarity). Specifically, after 12 h of visible light irradiation, $H_2O_2$ yield by COF-N32 reaches 7092 μmol g$^{-1}$ (605 μmol g$^{-1}$ h$^{-1}$), which is greatly higher than that by COF-N31 (4316 μmol g$^{-1}$, 442 μmol g$^{-1}$ h$^{-1}$) and COF-N33 (1736 μmol g$^{-1}$, 155 μmol g$^{-1}$ h$^{-1}$) (Figs. 3a and S22). Negligible amount of $H_2O_2$ (<5%) could be degraded by metal-free COF-N32 under visible light irradiation (Fig. S23), indicating the inhibited side reaction. This contributes to the stable and high yield of $H_2O_2$ in water by COF-N32. $H_2O_2$ yields by COF-N32 is also higher than those by conventional photocatalysts including $TiO_2$, g-$C_3N_4$ and $WO_3$ (Figs. 3a and S22). Moreover, COF-N32 can yield over 3.17 mmol g$^{-1}$ h$^{-1}$ with the addition of 1 mg COF-N32 in 50 mL ultrapure water after 3 h of visible light irradiation (Fig. S24), which is much higher than those of recently reported photocatalysts in pure water under similar measurement conditions (Fig. 3b and Table S3). In addition, COF-N32 exhibits high apparent quantum yield (AQY) of 6.2% at 459 nm (Fig. 3c). The solar-to-chemical efficiency of COF-N32 (0.31%) under visible light irradiation (details are provided in Supporting information) is greatly higher than solar-to-biomass efficiency by plants (-0.1%).

Due to the inhibited side reaction, COF-N32 also exhibits stable $H_2O_2$ yield and excellent reusability for 5 reused cycles (Fig. 3d). No obvious structural change is observed in COF-N32 after use (Fig. S25), indicating its excellent photo-stability under visible light irradiation. The stable $H_2O_2$ yield during the 5 reused cycles and no obvious change of crystalline structure after use are also achieved for COF-N33

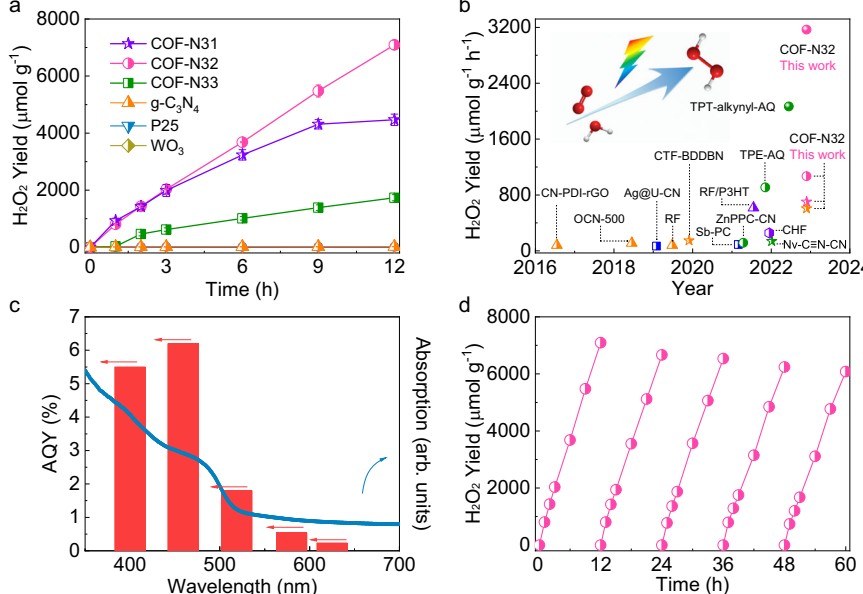

**Fig. 3 | Photocatalytic performance. a** Photosynthesis of $H_2O_2$ by COF-N31, COF-N32 and COF-N33. Experimental conditions: $\lambda > 420$ nm (298 K; xenon lamp, light intensity:100 mW $cm^{-2}$), ultrapure water (50 mL), photocatalyst (25 mg). **b** The comparison of $H_2O_2$ photosynthesis rate by COF-N32 with other reported photocatalysts without sacrificial agent under the similar measurement conditions (shape of symbol refers to the dosage of photocatalysts, sphere: 1 mg; circle: 10 mg; star: 20–30 mg; triangle: 50 mg; square: 100 mg; hexagon: 375 mg; color of symbol refers to the reaction duration, green: 1 h; blue: 1.2–2 h; pink: 3 h; purple: 6 h; orange: 10–24 h). **c** The apparent quantum yield (AQY) of COF-N32 as a function of wavelength (purple light at 400 nm, blue light at 459 nm, green light at 519 nm, yellow light at 580 nm, red light at 625 nm). **d** The reusability of COF-N32 for $H_2O_2$ photosynthesis. Experimental conditions: $\lambda > 420$ nm (298 K; xenon lamp, light intensity:100 mW $cm^{-2}$), ultrapure water (50 mL), photocatalyst (25 mg). Error bars in **a** represent the average values (mean ± s.d., $n = 3$).

(Figs. S26a and S26b). In contrast, the $H_2O_2$ yield by COF-N31 dramatically decreases with increasing reused cycles (Fig. S26c), indicating the relatively low photo-stability of COF-N31 during the reaction duration. The decreased crystallinity of COF-N31 after photocatalytic reaction (Fig. S26d) confirms its low photo-stability during the reaction process. The photo-stability observed for both COF-N32 and COF-N33 during the reaction duration indicates that $h^+$ generated by COF-N32 and COF-N33 directly oxidizes water to $H_2O_2$ instead of attacking COFs. The WOR directly to $H_2O_2$ by COF-N32 and COF-N33 (thermodynamically feasible stated above, Fig. S21) is confirmed by isotopic experiments using $H_2^{18}O$ and $^{16}O_2$ as precursors in a sealed reactor (Figs. S27b and S27c). Unlike that obtained for COF-N32 and COF-N33 (two COFs with relatively weak intramolecular polarity), WOR directly to $H_2O_2$ by COF-N31 with strong intramolecular polarity yet is not thermodynamically feasible (Fig. S21), which is also confirmed by isotopic experiments (Fig. S27a). Instead, $h^+$ generated by COF-N31 can attack COF itself, leading to the low photo-stability of COF-N31 during the reaction duration. Note the quench of $h^+$ by COF-N31 can promote the separation of $e^- -h^+$ pairs, which is beneficial for the $H_2O_2$ production[44,45]. Therefore, even though the amount of free charges (i.e. $e^-$ and $h^+$) by COF-N31 is much lower than that by COF-N32 (Figs. S18a and 2d), $H_2O_2$ production rate by COF-N31 is similar to that by COF-N32 at the beginning of photocatalytic process (Fig. 3a). The decreased $H_2O_2$ production rate observed with the increasing reaction duration can be attributed to the self-decomposition of COF-N31 during the reaction duration, which is also confirmed by the decreased crystallinity of COF-N31 after photocatalytic reaction (Fig. S26d). Similar observation about self-decomposition of COFs during $H_2O_2$ photosynthesis in water has also been reported previously[46]. The results clearly show that in pure water, COF-N31 with strong intramolecular polarity has low photo-stability of COF-N31 during the reaction duration, whereas COF-N32 and COF-N33 with relatively weak intramolecular polarity especially COF-N32 owns excellent photo-stability under visible light irradiation and can be consecutively reused for the photo-generation of $H_2O_2$.

The $H_2O_2$ yield (~3.5 mM) by COF-N32 after 12 h with visible light irradiation (Fig. 3a) can meet the $H_2O_2$ concentration required for water purification[47]. The filtrate of $H_2O_2$ generated by COF-N32 can be directly used to efficiently inactivate antibiotic resistant bacteria under dark condition, indicating that the produced $H_2O_2$ can be employed for water disinfection (Fig. S28). In addition to ex-situ disinfection, COF-N32 can also in-situ disinfect antibiotic resistant bacteria and degrade organic pollutant with emerging concerns (diclofenac) under visible light irradiation (Fig. S29). More importantly, COF-N32 is able to produce $H_2O_2$ at wide ranges of initial solution pH (3–11) (Fig. S30). Due to the consumption of radicals by dissolving ions and natural organic matter (NOM), the $H_2O_2$ production in real waters has been previously found to be inhibited[48,49]. However, we experimentally demonstrate that the efficient $H_2O_2$ photosynthesis by COF-N32 can also be achieved in more available real water samples with complex water matrix conditions. The $H_2O_2$ photosynthesis rate by COF-N32 reaches 667, 648 and 554 µmol $g^{-1}$ $h^{-1}$ in tap water, river water and sea water within 3 h of visible light irradiation, respectively (Fig. S31). The $H_2O_2$ photosynthesis in a cheap commercial membrane filter reactor (facile for separation of COF-N32 after use) was also investigated under simulated visible light irradiation. Regardless without or with using rubber plug to prevent the penetration of water through membrane in reactor under gravity, COF-N32 can efficiently produce ~30 µmol $H_2O_2$ in 2 h for the successive four cycles (Fig. S32).

The $H_2O_2$ photosynthesis by COF-N32 was further examined under natural sunlight irradiation (Figs. 4a–d). In two different reactor systems (double-walled beaker and membrane filter reactor), COF-N32 can efficiently produce $H_2O_2$ in different types of water samples under natural sunlight both in cloudy and sunny days (Figs. 4a, c, d, and S33–35). Although the used COF-N32 can be easily recovered by membrane filtration in double-walled beaker or can be separated in membrane filter reactor with the removal of $H_2O_2$ solution under gravity, COF-N32 powders were immobilized onto indium tin oxide (ITO) glass slide to further ease its recovery and reuse for practical

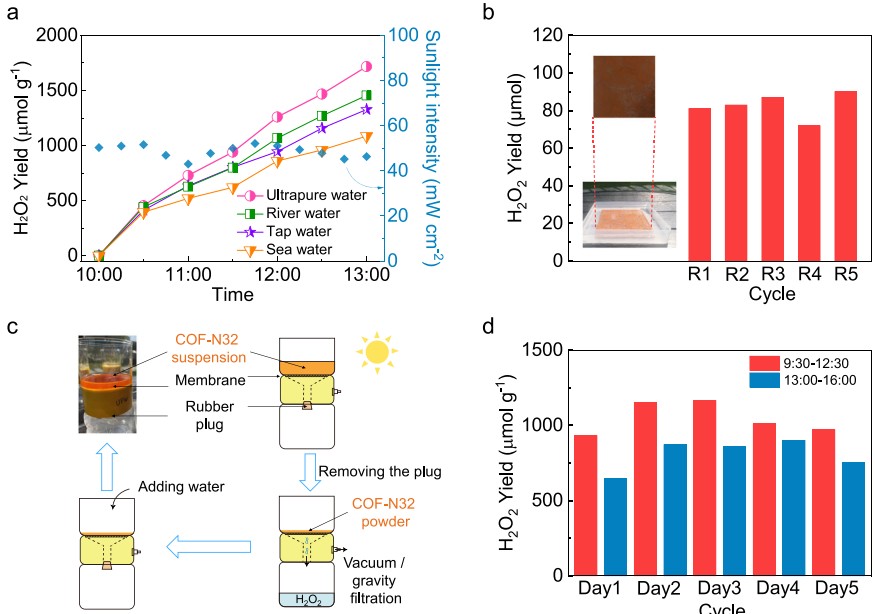

**Fig. 4 | Feasibility investigation for practical applications. a** Photocatalytic $H_2O_2$ production by COF-N32 in 50 mL of ultrapure water, tap water, river water and sea water under natural sunlight irradiation on a sunny day. **b** Immobilization of COF-N32 powders (60 mg) onto ITO glass slide (10 cm × 10 cm) for $H_2O_2$ production in 200 mL of pure water under natural sunlight irradiation from 10:30 to 13:30 on sunny days. **c** The schematic illustration of $H_2O_2$ photosynthesis process in a membrane reactor. **d** The photocatalytic production of $H_2O_2$ by COF-N32 in a membrane reactor under natural sunlight irradiation. Experimental conditions: ultrapure water (50 mL), photocatalyst (25 mg).

application (Fig. 4b). Pre-experiments show that ~3.5 µmol of $H_2O_2$ can be produced in every 2 h for the consecutive five cycles by COF-N32-immobilized glass slide (2 cm × 2 cm, containing 5 mg COF-N32) under visible light irradiation (Fig. S36). In a scaled-up reactor with working volume of 200 mL, ~85 µmol of $H_2O_2$ can be generated in 3 h on five sunny days by a larger COF-N32-loaded glass side (10 cm × 10 cm glass slide with the loading of 60 mg COF-N32) under natural sunlight irradiation (Fig. 4b). The results suggest the feasibility of panel-level $H_2O_2$ photosynthesis based on COFs photocatalysts.

## Photocatalytic mechanisms of COFs

The generation process of $H_2O_2$ by COF-N32 under visible light irradiation was investigated by quenching experiments and in-situ ESR analysis. Negligible $H_2O_2$ is generated within 12 h when dissolved $O_2$ in water is eliminated by $N_2$ purging (Figs. 5a and S37), indicating $O_2$ is essential for $H_2O_2$ production. Rotating ring-disk electrode (RRDE) analysis shows that the number of electrons transferred from COF-N32 to $O_2$ is estimated to be 2.17 (Fig. S38), indicating that $O_2$ is generally reduced to generate $H_2O_2$ via the apparent 2-electron reaction. The intermediates during oxygen reduction were further investigated by the trapping experiments. The addition of *p*-benzo-quinone (*p*-BQ, the scavenger of $·O_2^-$) into reaction system significantly inhibits the production of $H_2O_2$ with negligible yield, suggesting $·O_2^-$ is an intermediate product crucial for the $H_2O_2$ photosynthesis. Note that $·O_2^-$ is the reduction product of $O_2$ by electrons ($e^-$) photo-generated by COFs (Eq. S5). As stated above, among three COFs, the amount of $e^-$ generated by COF-N32 is the largest (Fig. 2d). The DMPO-$·O_2^-$ intensity in the in-situ ESR spectra for COF-N32 thus is higher than the other two COFs (Fig. S39). This indicates that regardless the amount of $O_2$ adsorbed by COF-N32 was not the highest among three COFs (Figs. S40 and S41), the electron transfer efficiency from COF-N32 to $O_2$ for the production of $·O_2^-$ species was higher than the other two COFs. Accordingly, the $H_2O_2$ yield by COF-N32 is the highest among three COFs. As mentioned above, $e^-$ generated from the dissociation of formed excitons by COFs with light irradiation can react with $O_2$ to produce $·O_2^-$ and subsequently to

$H_2O_2$ (Eqs. S5 and S6). $h^+$ generated from the separation of $e^-$-$h^+$ pairs (excitons) yet can oxidize $H_2O$ to form adsorbed *OH (Eq. S7) and then to $H_2O_2$ (Eq. S8). Consistently, diffusing ·OH is not detected by ESR in COF-N32 reaction system (Fig. S42), suggests the fast oxidation process of *OH to $H_2O_2$. Meanwhile, the introduction of tertiary butanol (TBA) has negligible effect on $H_2O_2$ yield by COF-N32 ($p > 0.1$), indicating that diffusing ·OH does not have contribution to the photocatalytic process of $H_2O_2$ production.

In-situ FTIR spectra in sealed chamber was further employed to reveal the reactive sites on COF-N32 during the $H_2O_2$ generation process. The results show that $H_2O$ adsorbed onto surfaces of COFs without light irradiation can be dissociated into C-OH$^-$ (1094 cm$^{-1}$) and TzH$^+$ (1508 cm$^{-1}$) in triazine rings on COF-N32 (Fig. S43)[50]. To further elucidate the water oxidation process by COF-N32 with light irradiation, the in-situ FTIR spectra with the presence of $O_2$ and $H_2O$ was also achieved. With the increase of reaction duration, both peaks at 1182 cm$^{-1}$ corresponding to N-O and 1379 cm$^{-1}$ corresponding to O-H bonds[51] in COF-N32 are found to be increased. Moreover, TzH$^+$ (at 1522–1557 cm$^{-1}$) is also generated (Figs. 5b and S44). The observation indicates that the water oxidation surrounding the N atoms in triazine rings leads to the formation of adsorbed *OH and H$^+$. This allows the occurrence of water oxidation with relatively low overpotential of VB compared with E($H_2O_2$/$H_2O$) (1.77 V vs. NHE)[52]. The production of $H_2O_2$ in half-reaction (with the removal of $e^-$ by NaBrO$_3$) under the $N_2$ atmosphere (Fig. S45) confirms the presence of two-electron water oxidation process in COF-N32 reaction system, while the negligible $O_2$ generation in oxygen evolution experiment excluded four-electron water oxidation (Fig. S46). The isotopic experiment (using $H_2^{18}O$ and $^{16}O_2$ as precursors in a sealed reactor) shows that the amount of $^{16}O_2$ and $^{18}O_2$ generated from the decomposition of $H_2O_2$ is generally equivalent (Fig. S27b), which further confirms the presence of two-electron water oxidation as well as the charge conservation with oxygen reduction during the $H_2O_2$ production process. It is worth pointing out that replacing triazine rings in COF-N32 by benzene rings (COF-C32) can decrease the yield of $H_2O_2$ under visible light irradiation (Fig. S47). The observation indicates that introduction of triazine

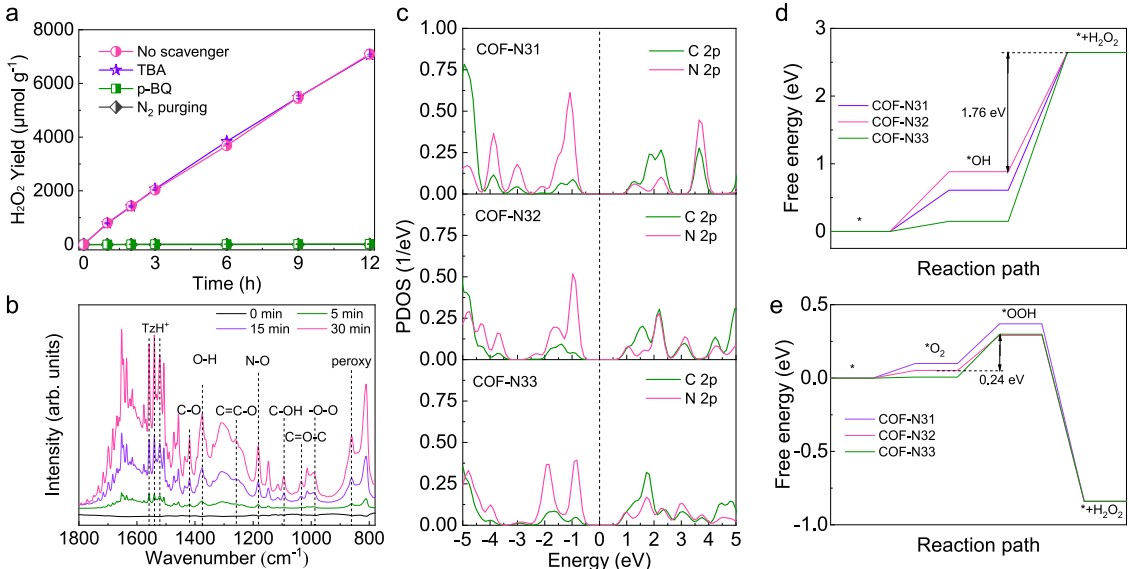

**Fig. 5 | Photocatalytic mechanisms for $H_2O_2$ production process. a** Quenching experiments for $H_2O_2$ photosynthesis. Experimental conditions: λ > 420 nm (298 K; xenon lamp, light intensity:100 mW cm$^{-2}$), volume (50 mL), photocatalyst (25 mg), [$p$-BQ]$_0$ = 3 mM, [TBA]$_0$ = 3 mM. **b** Time-course in-situ FTIR spectra of $O_2$ on COF-N32 under visible light irradiation with $O_2$. **c** PDOS of COF-N31, COF-N32 and COF-N33. The dashed lines stand for the Fermi level. Calculated energy profile for (**d**) oxidation of water into $H_2O_2$ and (**e**) reduction of oxygen into $H_2O_2$ on COF-N31, COF-N32 and COF-N33 at U = 0 V vs. SHE at pH = 7. Error bars in **a** represent the average values (mean ± s.d., $n$ = 3).

structure into COFs is favorable for $H_2O_2$ production via facilitating water oxidation. The vibration peak at 990 cm$^{-1}$ is corresponded to adsorbed ·$O_2^-$ and that at 861 cm$^{-1}$ is corresponded to peroxy species (-O-O-) in the in-situ FTIR spectra of COF-N32 (Figs. 5b and S48)[53]. The formation of C-O (1419 cm$^{-1}$) and C = C-O (1264 cm$^{-1}$) in the in-situ FTIR spectra of COF-N32 further suggest the reduction of $O_2$ occurs at C atoms of COF-N32.

The $H_2O_2$ photosynthesis process by COFs was further revealed by theoretical calculation. The calculated PDOS plots show that the valence band top and conduction band bottom in all three COFs mainly consist of N $2p$ and C $2p$ states, respectively (Fig. 5c). This implies that during the photocatalytic reaction in all three COFs, the N sites act as the oxidizing centers, while the C sites serve as reducing centers. Comparing with COF-N31 with weakened π conjugation, the N $2p$ states in COF-N33 move toward a more negative region. N $2p$ states of COF-N32 locate between those of COF-N31 and COF-N33. Note that more negative N $2p$ states can easily lead to the formation of unstable chemical bond between catalyst and reactant molecule. In contrast, more positive N $2p$ states readily causes the formation of inert chemical bond, which is not beneficial for the catalytic reaction either[54]. Instead, a moderate location of N $2p$ can effectively improve the photocatalytic activity. Likewise, the location of C $2p$ states in COF-N32 is also between those in COF-N31 and COF-N33. Hence, the $H_2O_2$ yield by COF-N32 is greatest among three COFs. The energy profiles of $H_2O_2$ production by three COFs were further determined. As shown in Fig. 5d, e, the water oxidation is the rate-determining reaction in the photocatalytic water and oxygen reforming. Moreover, the conversion of *OH into *$+H_2O_2$ is the rate-determining step in water oxidation[55]. Specifically, COF-N32 owns a lowest energy barrier (1.76 eV) among three COFs. Meanwhile, the energy barrier of oxygen reduction into $H_2O_2$ by COF-N32 is only 0.24 eV, which is also the lowest among COFs. Hence, the highest $H_2O_2$ yield is obtained by COF-N32 with light irradiation.

In summary, in order to enhance the yield of $H_2O_2$ by COFs under light irradiation, we propose a strategy to facilitate the formation and dissociation of excitons in COFs through optimizing the intramolecular polarity of COFs by introducing suitable amount of phenyl group as electron donors. We fabricate a class of COFs with different intramolecular polarity by using triazine-cored triamine with different

amount of phenyl group ($n$ = 0, 1, 2) as the precursors. We find that among all three COFs, COF-N32 with moderate intramolecular polarity in the octupolar conjugated structure can generate greatest amount of charges (i.e. electrons) for $H_2O_2$ photosynthesis under light irradiation. Without the requirement of sacrificial agent, the $H_2O_2$ yield by COF-N32 reaches 7092 μmol g$^{-1}$ (605 μmol g$^{-1}$ h$^{-1}$) after 12 h of visible light irradiation with solar-to-chemical efficiency of 0.31% and high AQY of 6.2% at 459 nm. Moreover, we find that COF-N32 can also efficiently produce $H_2O_2$ in more available real water samples including tap water, river water and sea water. COF-N32 can be assembled into practical devices for consecutive uses with high photocatalytic stability. COF-N32 either dispersed in membrane filter reactors or immobilized onto glass slides can efficiently produce $H_2O_2$ under natural sunlight irradiation. During the $H_2O_2$ photosynthesis process, suitable N $2p$ states and C $2p$ states in COF-N32 with optimal intramolecular polarity reduce the energy barrier for $H_2O$ activation and oxygen reduction, respectively, contributing to the high efficiency. This study not only provides deep insight into the design of COFs via regulating its intramolecular polarity to boost the $H_2O_2$ photosynthesis without using sacrificial agent, but also paves the way for the practical application of COFs-based photosynthesis of $H_2O_2$.

## Methods
### COFs synthesis
Unlike COF-N32 and COF-N33 that can be fabricated by using one-pot solvothermal method based on the Schiff-base reaction between two types of precursors in mesitylene/dioxane/acetic acid solvent, COF-N31 yet can not be successfully synthesized by the mixture of 1,3,5-triformylphloroglucinol (Tp) and melamine in mesitylene/dioxane/acetic acid solvent due to its instability in this solvent. Instead, COF-N31 were fabricated in dimethyl sulfoxide/N,N-dimethylacetamide (DMAc)/acetic acid following the method reported in previous literature[22]. Specifically, 0.3 mmol of Tp (63 mg) and 0.3 mmol of melamine (38 mg) were added into a reactor, followed by the addition of 2 mL of dimethyl sulfoxide, 1 mL of DMAc and 0.3 mL of 6 M acetic acid. After ultrasonication and degassed by three consecutive freeze-pump-thaw cycles, the reactor was sealed under vacuum condition,

which was then heated at 120 °C for 72 h. The collected product was firstly rinsed by DMAc, which was then solvent exchanged with DMAc, pure water and washed with acetone for three times. The final product was dried at 120 °C under vacuum.

COF-N32 and COF-N33 were synthesized by mixing Tp with the precursors with triazine and triamine in mesitylene/dioxane/acetic acid solvent, respectively. Briefly, 0.9 mmol (190 mg) of Tp and 0.9 mmol of 4,4',4''-(1,3,5-triazine-2,4,6-triyl)-trianiline (318.6 mg, for COF-N32) or 4,4',4''-(1,3,5-triazine-2,4,6-triyl)tris(([1,1'-biphenyl]−4-amine)) (524 mg, for COF-N33) were added into a Teflon lining with the volume of 20 mL. Then, dioxane (4.5 mL), mesitylene (4.5 mL) and 3 M acetic acid (1.5 mL) were added. The mixture was ultrasonicated and bubbled with $N_2$ for 20 min. After that, the Teflon lining was sealed in an autoclave and heated at 120 °C in the oven for 72 h. The mixture was separated through filtration and the solid was rinsed by acetone for five times, which was finally dried at 60 °C. The precursors-to-COFs efficiency is 76% for COF-N31, 78% for COF-N32 and 75% for COF-N33, respectively.

### Photocatalytic experiments

The photosynthesis of $H_2O_2$ was first performed in a double wall quartz reactor with 50 mL of ultrapure water and 25 mg of photocatalysts (1, 5 and 10 mg of COF-N32 were also considered in particular experiments). The initial solution pH of the mixture was adjusted by NaOH or $HClO_4$. The reaction suspension was irradiated by using a 300 W Xenon lamp ($\lambda > 420$ nm, $100 \pm 1$ mW cm$^{-2}$). The temperature was fixed at $25.0 \pm 0.2$ °C by circulating water system during the photocatalytic experiments. The reaction suspension was extracted and filtered for $H_2O_2$ measurement at specific time intervals. The reusability of three COFs was investigated. The $H_2O_2$ production by COF-N32 in various real waters (e.g. tap water, river water and sea water) was evaluated under both simulated visible light and natural solar light. COF-N32 were employed in commercial membrane reactors and further immobilized onto ITO glass with the size of 10 cm × 10 cm for the continuous $H_2O_2$ photosynthesis.

### Catalyst characterization

XRD (DMAX-2400, Rigaku, Japan), XPS (Axis Ultra, Kratos, UK), SEM (JSM-F100, JEOL, Japan), TEM (Tecnai F30, USA), $^{13}$C NMR (Bruker-400 AVANCE III, Bruker, Switzerland), FTIR (Nicolet is50, Thermo Fisher, USA) were employed to reveal the chemical and structural information of COFs. In-situ FTIR spectra measurement (Bruker Tensor, Bruker, Switzerland), UV-vis DRS (UV-2400, Shimadzu, Japan), steady state PL spectra, time-resolved PL decay curve, temperature-dependent PL spectra (FLS980, Edinburgh, UK), TA spectrometer (Helios, Ultrafast System, USA), ESR analysis (Bruker EMX, Bruker, Switzerland), water adsorption analysis (3Flex, Micromeritics, USA), RRDE (PINE E6, USA), $^{18}$O isotopic experiment and EIS (CHI760E, Chenhua, China) were performed to investigate the mechanisms of $H_2O_2$ photosynthesis by prepared COFs.

## Data availability

The data that support the findings of this study are available within the article and its Supplementary Information. Source data are provided with this paper.

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

## Acknowledgements

This work was supported by Fund for Innovative Research Group of NSFC under Grant No. 51721006 and the National Natural Science Foundation of China under Grant No. 42025706 and 52270015.

## Author contributions

M.P.T. and F.Y.L designed the research. F.Y.L. and P.Z. performed the research with the help of Y.H.H., H.T., and Y.L. F.Y.L., P.Z. and M.P.T. wrote the paper. J.L.L, Q.Z. S.J.G and J.R.N. provided ideas. All the co- authors contributed to interpretation of the findings.

## Competing interests

The authors declare no competing interests.
