## [Peer Review File · Nature Communications]

Covalent organic frameworks for direct photosynthesis of hydrogen peroxide from water, air and sunlightREVIEWER COMMENTS

Reviewer #1 (Remarks to the Author):

In this manuscript, the authors studied the relationship between photocatalytic performance and intramolecular polarity. They designed a series of COFs materials by using suitable amounts of phenyl groups as electron donors, that can maximize the free charge generation, leading to the record-high photocatalytic H₂O₂ synthesis (702 $\mu\text{mol g}^{-1} \text{h}^{-1}$) from water, oxygen and visible light without requiring a sacrificial agent. They claim that the weak intramolecular polarity in D-A COFs constrains excitons dissociation, yet too strong intramolecular polarity inhibits excitons formation via a weakened π -conjugated effect. The concept is novel and interesting. I personally very like this idea. However, the three COFs have all been reported previously. More importantly, the structure analysis of COF-N31 looks problematic. The authors should address the questions below before publishing. A major revision is needed.

1. The main concern is the structure of COF-N31. The PXRD is unsatisfactory. I highly suggest the authors read this literature (J. Am. Chem. Soc. 2019, 141, 6152–6156) in which an exactly the same structure with COF-N31 was reported. The PXRD patterns look very different. The PXRD of the COF-N32 has a similar problem. The simulations and Pawley refinements of all three COFs should be done. The very related literature should be cited (J. Am. Chem. Soc. 2019, 141, 6152–6156; J. Am. Chem. Soc. 2017, 139, 13083–13091). Photocatalysis is a very complicated system that involves many factors. The crystallinity and porosity of the obtained COFs may also have significant influence on the catalytic activities. I would suggest improving the crystallinity and porosity of COF-N31 first before testing its photocatalytic activity.

2. The BET surface area of COF-N31 (81 m²/g) is much lower than COF-N32 (823 m²/g) and COF-N33 (677 m²/g). The standard of the materials is different for the three COFs. Although the author claimed that “The surface area and O₂ absorption of COFs do not have close correlation with their H₂O₂ production.” The reason and evidence should be provided.

3. TEM images of the three COFs should be provided.

4. The porous properties can not be seen from Scanning electron microscopy (SEM) images.

5. I would suggest to label the names of the three COFs in Figure 1a.

Reviewer #2 (Remarks to the Author):

The manuscript entitled “Engineering intramolecular polarity of covalent organic frameworks for boosting direct photosynthesis of hydrogen peroxide from water, air and sunlight” provide a strategy for the design of a novel donor-acceptor (D-A)-type COFs with various intramolecular polarity by using triazine-cored triamine with different amount of phenyl group ($n = 0, 1, 2$) as the precursors. The COF-N32 with moderate intramolecular polarity in the octupolar conjugated structure can maximize the free charge generation, leading to the record-high photocatalytic H₂O₂ synthesis from water, oxygen and visible light without requiring sacrificial agent. The COFs have been studied in detail experimentally and theoretically for their optoelectronic and photocatalytic properties, but some of the explanations were not clear enough. For this, I recommend this manuscript be published in Nature Communications with major revision, after considering the comments below.

1. X-ray diffraction (XRD) patterns of COF-N31 is not reasonable, because XRD peaks

usually does not emerge after 10°.

2. As the VB level of COF-N32 is 1.88 V vs NHE, the overpotential for water oxidation is only <0.4 V, so the water oxidation to H₂O₂ seems to be difficult, and not well discussed in the current manuscript.

3. H₂O₂ production via two-electron water oxidation is an exciting work. But lack detail experimental evidences and not well discussed in the current manuscript.

4. Taking into account that the photo-generated holes could oxidize water to form hydroxyl radicals, could the formation of hydroxyl radicals affect the H₂O₂ production reaction?

5. In order to make the manuscript more general, the authors need to include these recent references. ACS Catal. 2022, 12, 12954-12963; Chemical Engineering Journal 454 (2023) 139929.

Reviewer #3 (Remarks to the Author):

The authors synthesized a class of D-A COFs with different intramolecular polarity by introducing suitable amounts of phenyl groups as electron donors for excitonic regulation to boost the direct photocatalytic H₂O₂ production from water, air, and sunlight without using sacrificial agent. The optimal COF-N32 can facilitate excitons formation and dissociation, leading to the record-high H₂O₂ yield (702 μmol g⁻¹ h⁻¹) with SCC of 0.31%. In addition, COF-N32 can also be verified the potential application feasibility with high photocatalytic stability. H₂O₂ photosynthesis is hot topic, and the photocatalytic performance reported in this manuscript is impressive. As a result, this reviewer would like to recommend the acceptance of this manuscript for publication in Nat. Commun. after clarifying the following issues.

1. In the paragraph describing “Photocatalytic H₂O₂ production by COFs”, authors try to convince the readers of the best photocatalytic performance of COF-N32. They said those “H₂O₂ yield by COF-N32 reaches 702 μmol g⁻¹ h⁻¹” and “which is much higher than those of recently reported photocatalysts in pure water under same measurement conditions (Figures 3b and S22, Table S3).” However, the given H₂O₂ yield was 3168 μmol g⁻¹ h⁻¹ in Figure 3b, which is taken from that in Table S3 and doesn't appear even once in the text. The amount of COF-N32 seems to lead to different yields, however there is no corresponding description in the text. Explanation should be provided for the graphic and text discrepancies.

2. The morphology of COFs is suggested to be further characterized by Transmission electron microscopy.

3. Rotating ring-disk electrode measurements are suggested to be carried out to explore the number of electron transferred during the catalytic reaction.

4. The standardization of various abbreviations and spelling should be carefully checked. For example, “COF-32” in lines 28 and 242 would be corrected to “COF-N32”.

Reviewer #4 (Remarks to the Author):

In this paper, Tong et al. report a strategy to regulate the intramolecular polarity of COFs to enhance the generation and separation of photoinduced excitons, thus optimizing the performance of photocatalytic H₂O₂ generation by these COFs. They claimed that a COF with a moderate intramolecular polarity that contains a triazine core and two peripheral benzenes exhibited the greatest performance for H₂O₂ production. The generalizability of this COF photocatalyst was verified by varying the water source, assembling it into practical devices and reusing for several times, which gives satisfactory performance in all the cases.

The authors investigated the mechanism of H₂O₂ production by conducting quenching experiments, in-situ ESR analysis, in-situ FTIR analysis and DFT calculations, elucidating the simultaneous oxygen reduction and water oxidation pathways and revealing the triazine nitrogen and carbon as the active sites for water oxidation and O₂ reduction, respectively. In summary, a comprehensive structure-property-function relationship was established in this paper, which provides a clear clue for the design of COFs via regulating their intramolecular polarity to boost photocatalytic performance. The whole manuscript is arranged logically and well-written. The figures and illustrations are appropriate for interpretation. The conclusion was well supported by the experimental data and analysis. Thus, I think this manuscript meets the criteria of Nature Communications and I recommend the publication of this manuscript after addressing the following concerns.

1, the author claimed the record-high H₂O₂ yield (702 μmol g⁻¹ h⁻¹) for a COF photocatalyst in the absence of a sacrificial reagent. From the viewpoint of charge conservation, theoretically, the electrons generated on COF which reduce oxygen should come from the oxidation of water. In order to exclude the self-oxidation of COF itself, H₂O₁₈ is suggested to use to verify the water oxidation process. The amount of 2 * 18O₂ and H₂18O₂ in the system should be equal to the H₂16O₂ to meet the requirement of charge conservation. Otherwise, self-oxidation of COF cannot be excluded.

2, "The signal intensities of three COFs follow the order of COF-N31 > COF-N32 > COF-N33 (Figure S5), which might result from the difference in intramolecular polarity among three COFs". As we know, the ESR signal comes from unpaired electrons in the materials. How do the authors draw a conclusion that different ESR signal intensities result from the different polarities in COFs? Is there any relationship between unpaired electrons and intramolecular polarity?

3, I doubt the relationship between water contact angles and polarity. Because measurement of water contact angles was in macroscale and the polarity was usually described in molecular scale. Also, the measurement of water contact angles will likely be influenced by many factors such as the preparation of samples, material morphology, temperatures, humidity, and so on. Alternatively, water adsorption analysis is suggested to study the polarity because the adsorption usually takes place in micro to mesoscale, which might be more convincing.

4, "Among three COFs, COF-N32 exhibits smaller average lifetime (86 ps) relative to COF-N31..." what kind of species do the lifetimes represent? photoexcited electrons? Holes? Or excitons?

5, The author performed the isopropanol (IPA) quenching experiment to trap the diffusing ·OH. However, isopropanol might act as a sacrificial agent which directly extracts holes. So is this appropriate to use IPA as the trapping agent of diffusing ·OH?

6, In Figure 2a, two transitions can be observed for COF-N33, which is different from COF-N31 and N33. How to explain this as the three COFs are anal

Reviewer #5 (Remarks to the Author):

In this work, the authors have synthesized and characterized a set of three covalent organic frameworks (COFs) that show a cost-effective approach to producing H₂O₂ from the air, water, and sunlight. In addition, they present a strategy for designing metal-free COFs with optimal properties to produce H₂O₂. The work is interesting, but there is still some additional work that needs to be done before its suitable for publication. As a result, I cannot recommend it for publication at the moment. My detailed comments are provided below:

1. The authors need to be explicit in the text about how they quantified the H₂O₂ that was produced. It was sort of frustrating to dig for this vital information.
2. The authors should check for the production of hydrogen and oxygen since the band gap suggests this material should be capable of oxidizing water to O₂ and H₂.
3. The authors should refrain from using SEM to qualify the porous nature of these materials.

SEM would describe the morphology of the material. If the authors are adamant about using an imaging technique to report the pore size of the COFs, suggest they obtain high-resolution TEM. Surprisingly, the author uses an imaging technique to qualify the pore size distribution of the material when the acceptable standard is to use BET analysis. Since the authors have already conducted BET measurements for these COFs to quantify the surface area of these COFs, they should report the pore-size distribution from the BET analysis.

4. The authors should obtain a better PXRD profile. I suggest using a Voltage of 30 and a Current of 20 and running the experiment for at least 40 minutes. Running PXRD with high power induces fluoresces in the material, resulting in poor peaks.

5. The authors should obtain simulated PXRDs of the COFs to determine the interlayer stacking of these materials. This information is crucial because the authors claim that these are novel COFs. Additionally, determining the interlayer stacking will also help in explaining their results.

Response to Reviewer 1 :

Overall comments: In this manuscript, the authors studied the relationship between photocatalytic performance and intramolecular polarity. They designed a series of COFs materials by using suitable amounts of phenyl groups as electron donors, that can maximize the free charge generation, leading to the record-high photocatalytic H₂O₂ synthesis (702 μmol g⁻¹ h⁻¹) from water, oxygen and visible light without requiring a sacrificial agent. They claim that the weak intramolecular polarity in D-A COFs constrains excitons dissociation, yet too strong intramolecular polarity inhibits excitons formation *via* a weakened π-conjugated effect. The concept is novel and interesting. I personally very like this idea. However, the three COFs have all been reported previously. More importantly, the structure analysis of COF-N31 looks problematic. The authors should address the questions below before publishing. A major revision is needed.

Response: Thanks so much for your great efforts in reviewing our revised manuscript. We sincerely appreciate your valuable comments and suggestions. We have revised the manuscript according to your valuable suggestions as well as those from other Reviewers. We believe that the revised manuscript has been substantially strengthened. We are looking forward to your continuous support for our revised manuscript.

Comment 1: The main concern is the structure of COF-N31. The PXRD is unsatisfactory. I highly suggest the authors read this literature (*J. Am. Chem. Soc.* 2019, 141, 6152–6156) in which an exactly the same structure with COF-N31 was reported. The PXRD patterns look very different. The PXRD of the COF-N32 has a similar problem. The simulations and Pawley refinements of all three COFs should be done. The very related literature should be cited (*J. Am. Chem. Soc.* 2019, 141, 6152–6156; *J. Am. Chem. Soc.* 2017, 139, 13083–13091). Photocatalysis is a very complicated system that involves many factors. The crystallinity and porosity of the obtained COFs may also have significant influence on the catalytic activities. I would suggest improving the crystallinity and porosity of COF-N31 first before testing its photocatalytic activity.

Response 1: Thank you for your valuable comment. Following your excellent suggestion, COF-N31-DMSO has been successfully fabricated in the mixture of DMSO, DMAc and 6M HAc, according to the related pioneer literature (*J. Am. Chem. Soc.*, 2019, 141, 6152–6156). The experimental PXRD pattern of COF-N31-DMSO is consistent with the simulation (**Figure S2a**), indicating that COF-N31-DMSO exhibits high crystallinity. Meanwhile, by using PXRD analysis with voltage of 30 kV and 20 mA (measurement condition provided by Reviewer #5), the quality of XRD pattern is improved. The PXRD patterns of COF-N32 and COF-N33 are also consistent with the simulation (**Figure S2c and d**), confirming the high crystallinity of COF-N32 and COF-N33.

The original COF-N31 was fabricated in the mixture of mesitylene, 1,4-dioxane and 3 M HAc solution by using the same methods as COF-N32 and COF-N33. Since its crystallinity was not very high, the XRD pattern of COF-N31 does not match well with the simulation. Similar observations have been reported in previous studies about COF-N31 fabrication by ball-milling (*Chem. Commun.*, 2019,55, 167-170) or solvothermal methods (*Chem. Commun.*, 2017,53, 9636-9639; *J. Chromatogr. A*, 2020, 1619, 460916).

To investigate the photocatalytic property of COF-N31-DMSO, UV-DRS (**Figure R1a**) and steady-state PL spectra (**Figure R1b**) have been conducted. COF-N31-DMSO also exhibit wide band gap (2.71 eV) and low PL intensity, suggesting that COF-N31-DMSO with high calculated intramolecular polarity exhibits low light absorption but high charge separation. This observation is consistent with that of the original COF-N31. Therefore, the conclusion about the influence of intramolecular polarity on the generation and separation of excitons is not significantly affected by the crystallinity of COF-N31.

Previous study shows that COF-N31-DMSO can be employed in organic reaction with good photo stability (*J. Am. Chem. Soc.*, 2019, 141, 6152–6156). However, in present study, we find that the photo-stability is relatively low in pure water during H₂O₂ photosynthesis without the addition of sacrificial agent. Due to the high intramolecular polarity and high internal strain in COF-N31-DMSO, the intramolecular electric field

intensity of C-N linkage may be high, leading to the high reactivity of self-decomposition in the C-N linkage (*Nat Commun.*, 2018, 9, 2998; *Chem. Soc. Rev.*, 2020, 49, 8469-8500). As a result, COF-N31-DMSO is relatively easy to self-decompose in water under light irradiation, leading to the relatively low photo-stability. Specifically, the H₂O₂ yield of COF-N31-DMSO decrease to only 77 $\mu\text{mol g}^{-1} \text{h}^{-1}$ in the sixth reused cycle (**Figure R2a**). Meanwhile, the gradual decrease in H₂O₂ yield is also observed within 12 h in water (**Figure R2b**).

In contrast, original COF-N31 with relatively low crystallinity compared with COF-N31-DMSO exhibits high photo-stability (**Figure R2c**) and can continuously produce H₂O₂ within 12 h under visible light irradiation in water (**Figure R2d**). Therefore, the increase in crystallinity can not improve the overall photocatalytic H₂O₂ production by COFs in water without the addition of sacrificial agent. Instead, the intrinsic chemical structure would significantly dominate the photocatalytic capability, which has been demonstrated in many previous studies about conjugated polymers (e.g. Yan et al., *Proc. Natl. Acad. Sci. U. S. A.*, 2022, 119, e2202913119; Ye et al., *Proc. Natl. Acad. Sci. U. S. A.*, 2021, 118, e2103964118). Therefore, original COF-N31 fabricated by using the same methods as COF-N32 and COF-N33 is used in our following control experiments.

The corresponding texts have been added in the revised manuscript (Please see **lines 69-71** and **Figure S2**). “Based on the simulation and Pawley refinement, the obvious peaks at $2\theta = 9.6^\circ$, 5.5° and 3.9° corresponding to (100) planes and $2\theta = 27^\circ$ corresponding to (002) planes in the X-ray diffraction (XRD) pattern (**Figure S2**) suggest that three COFs contain crystalline structure²⁰⁻²².”

Figure S2. Experimental and simulated X-ray diffraction (XRD) patterns of (a) COF-N31-DMSO, (b) COF-N31, (c) COF-N32 and (d) COF-N33. Pawley refinements show the good agreement between experimental and simulated eclipsed XRD patterns ($R_{wp} = 2.67\%$, $R_p = 2.10\%$ for COF-N31-DMSO, $R_{wp} = 3.27\%$, $R_p = 2.47\%$ for COF-N32, $R_{wp} = 3.49\%$, $R_p = 2.50\%$ for COF-N33). Since its crystallinity was not very high, the XRD pattern of COF-N31 does not match well with the simulation. Similar observations have been reported in previous studies (*Chem. Commun.*, 2019, 55, 167-170, *Chem. Commun.*, 2017, 53, 9636-9639). Preliminary experiments show that COF-N31-DMSO with high crystallinity yet exhibits low photo-stability but similar excitonic behavior with COF-N31 (data not shown). Therefore, COF-N31 with the same chemical structure as COF-N31-DMSO is thus used in the following experiments.

Figure R1. (a) Tauc plot of COF-N31-DMSO and (b) steady-state PL spectra of COF-N31-DMSO as well as COF-N31, COF-N32 and COF-N33.

Figure R2. (a) The reusability of (a) COF-N31-DMSO and (c) COF-N31 for H₂O₂ photosynthesis. Long-term H₂O₂ photosynthesis by COF-N31-DMSO (b) and COF-N31 (d). Conditions: $\lambda > 420$ nm (298K; xenon lamp, light intensity: $100 \text{ mW} \cdot \text{cm}^{-2}$), ultrapure water (50 mL), photocatalyst (25 mg).

Synthesis of COF-N31-DMSO

“COF-N31-DMSO was fabricated by a solvothermal method according to the literature (*J. Am. Chem. Soc.*, 2019, 141, 6152–6156). Specifically, 0.3 mmol of Tp and 0.3 mmol of melamine were added into a reactor, followed by the addition of 2 mL of dimethyl sulfoxide, 1 mL of *N,N*-dimethylacetamide (DMAc) and 0.3 mL of 6 M acetic acid. After ultrasonication and degassed by three consecutive freeze-pump-thaw cycles, the reactor was sealed under vacuum condition, which was then heated at 120 °C for 72 h. The precipitation was first rinsed by DMAc. Then, the product was solvent exchanged with DMAc, pure water and washed with acetone for three times. The product was dried at 120 °C under vacuum.”

Comment 2: The BET surface area of COF-N31 ($81 \text{ m}^2 \text{ g}^{-1}$) is much lower than COF-N32 ($823 \text{ m}^2 \text{ g}^{-1}$) and COF-N33 ($677 \text{ m}^2 \text{ g}^{-1}$). The standard of the materials is different for the three COFs. Although the author claimed that “The surface area and O_2 absorption of COFs do not have close correlation with their H_2O_2 production.” The reason and evidence should be provided.

Response 2: Thank you for pointing out our omission. After the careful consideration, the correlation of surface area and O_2 absorption with H_2O_2 can not be totally excluded. Based on the results of BET surface area, TPD- O_2 analysis and the production of 1,4-endoperoxide species and adsorbed $\cdot\text{O}_2^-$, it can be only concluded that the electron transfer efficiency in oxygen reduction is more important than surficial properties during H_2O_2 production process. The corresponding texts has been rewritten in the caption of **Figure S48**.

Figure S48. N_2 adsorption-desorption isotherms of (a) COF-N31, (b) COF-N32 and (c) COF-N33 and the corresponding pore size distribution (insets). (d) O_2 -TPD curves of three COFs. The BET surface area of COF-N31, COF-N32 and COF-N33 is determined to be $81 \text{ m}^2/\text{g}$, $823 \text{ m}^2/\text{g}$ and $677 \text{ m}^2/\text{g}$, respectively. The three COFs shows narrow pore size of 1-2 nm, indicating their micropore structure. The intensities of O_2 -TPD signals follow the order of COF-N31 > COF-N32 > COF-N33. It should be noted that the COFs are thermally stable at the tested temperature ($< 200^\circ\text{C}$, **Figure S47**), suggesting the signals in O_2 -TPD are not resulted from the decomposition of COFs. Therefore, the O_2 adsorption is mainly relevant to the structure of COFs instead of surface area. Besides, COF-N31 with more O_2 adsorption exhibits relatively low production capability of 1,4-endoperoxide species and adsorbed $\cdot\text{O}_2^-$ compared with that of COF-N32, further implying that the electron transfer efficiency in oxygen reduction is more important than surficial properties during H_2O_2 production process.

Comment 3: TEM images of the three COFs should be provided.

Response 3: Thank you for your valuable comment. Following your excellent suggestion, we have added the corresponding text and TEM images in the revised manuscript (**line 71-74**) and Supporting Information. “Scanning electron microscopy (SEM) and transmission electron microscopy (TEM) images show that the three COFs are tiny granular particles with diameter of 2~3 μm , which are assembled by numerous nanorods (**Figures S3 and S4**).”

Figure S4. TEM images of (a, b) COF-N31, (c, d) COF-N32 and (e, f) COF-N33. The diffraction fringes can not be observed in COF-N31 and COF-N32 due to the relatively low crystallinity of COFs compared with inorganic semiconductors (*J. Am. Chem. Soc.*, 2017, 139, 13083–13091; *J. Am. Chem. Soc.*, 2019, 141, 6152–6156; *Chem. Commun.*, 2019, 55, 167–170). In COF-N33 with high crystallinity, the diffraction fringes of (100) and (002) planes can be found in **Figure S4e** and **S4f**, respectively.

Comment 4: The porous properties can not be seen from Scanning electron microscopy (SEM) images.

Response 4: Thank you for your valuable comment. We agree the Reviewer that SEM could only show morphology, instead of porous structure. The BET measurement has been added to identify the porous properties of COF samples. The three COFs shows narrow pore size of 1-2 nm, indicating their micropore structure. The corresponding text has been rewritten in the revised manuscript (**lines 72-74 and Figure S48**). “Scanning electron microscopy (SEM) and transmission electron microscopy (TEM) images show that the three COFs are tiny granular particles with diameter of 2~3 μm , which are assembled by numerous nanorods (**Figures S3 and S4**).”

Comment 5: I would suggest to label the names of the three COFs in Figure 1a.

Response 5: Thank you for your valuable comment. Following your excellent suggestion, we have added the names of three COFs in **Figure 1a**.

Figure 1. Chemical structure. (a) Schematic illustration of synthesis process of COF-N31, COF-N32 and COF-N33. (b) ¹³C NMR spectra and (c) Fourier transformation infrared spectroscopy (FTIR) of COF-N31, COF-N32 and COF-N33. (d) The schematic illustration of the octupolar structure in COF-N31, COF-N32 and COF-N33. (e) DFT calculation on charge distribution of COF-N32 structure.

Response to Reviewer 2 :

Overall comments: The manuscript entitled “Engineering intramolecular polarity of covalent organic frameworks for boosting direct photosynthesis of hydrogen peroxide from water, air and sunlight” provide a strategy for the design of a novel donor-acceptor (D-A)-type COFs with various intramolecular polarity by using triazine-cored triamine with different amount of phenyl group ($n = 0, 1, 2$) as the precursors. The COF-N32 with moderate intramolecular polarity in the octupolar conjugated structure can maximize the free charge generation, leading to the record-high photocatalytic H_2O_2 synthesis from water, oxygen and visible light without requiring sacrificial agent. The COFs have been studied in detail experimentally and theoretically for their optoelectronic and photocatalytic properties, but some of the explanations were not clear enough. For this, I recommend this manuscript be published in Nature Communications with major revision, after considering the comments below.

Response: Thanks so much for your great efforts in reviewing our revised manuscript. We sincerely appreciate your valuable comments and suggestions. We have revised the manuscript according to your valuable suggestions as well as those from other Reviewers. We believe that the revised manuscript has been substantially strengthened. We are looking forward to your continuous support for our revised manuscript.

Comment 1: X-ray diffraction (XRD) patterns of COF-N31 is not reasonable, because XRD peaks usually does not emerge after 10° .

Response 1: Thank you for your valuable comment. According to the related literature (*J. Am. Chem. Soc.*, 2019, 141, 6152–6156), COF-N31-DMSO was also synthesized. The simulations and Pawley refinements of COFs have been further done. The results show that COF-N31-DMSO with high crystallinity has been successfully fabricated. The different XRD pattern of original COF-N31 may be attributed to its not very high crystallinity. Similar observation has also been reported in previous studies about COF-N31 fabrication by ball-milling (*Chem. Commun.*, 2019,55, 167-170) or solvothermal methods (*Chem. Commun.*, 2017,53, 9636-9639; *J. Chromatogr. A*, 2020, 1619, 460916).

Comment 2: As the VB level of COF-N32 is 1.88 V vs NHE, the overpotential for water oxidation is only <0.4 V, so the water oxidation to H_2O_2 seems to be difficult, and not well discussed in the current manuscript.

Response 2: Thank you for your valuable comment. According to the ^{18}O isotopic experiment, $\text{H}_2^{18}\text{O}_2$ was generated by using H_2^{18}O and $^{16}\text{O}_2$ in COF-N32 system, confirming the two-electron water oxidation by COF-N32 under visible light irradiation. According to the DFT calculation, the energy barrier (**Figure 5d**) of water oxidation is relatively high compared with oxygen reduction (**Figure 5e**). Therefore, water oxidation is the rate-determining step, rather than the O_2 reduction. The direct conversion of $^*\text{OH}$ to $^* + \text{H}_2\text{O}_2$ is a quick interfacial reaction, which is not relevant to the diffusion, leading to the low requirement of overpotentials in COF-N32 photocatalyst. Similar observations have been also reported in the previous studies about organic photocatalysts for two-electron water oxidation (*Adv. Mater.*, 2020, 32, e1904433; *Adv. Mater.*, 2022, 34, 2107480).

The corresponding explanations have been added into the revised manuscript (Please see **lines 240-241**). “This allows the occurrence of water oxidation with relatively low overpotential of VB compared with $E(\text{H}_2\text{O}_2/\text{H}_2\text{O})$ (1.77 V vs. NHE)⁴⁹.”

Comment 3: H_2O_2 production *via* two-electron water oxidation is an exciting work. But lack detail experimental evidences and not well discussed in the current manuscript.

Response 3: Thank you for your valuable comment. As referred in the Response 2, ^{18}O isotopic experiment was conducted to confirm the two-electron water oxidation. By using H_2^{18}O and $^{16}\text{O}_2$ as precursors, H_2O_2 was generated by COF-N32 under visible light irradiation. After purging N_2 to remove excessive $^{16}\text{O}_2$ and air, catalase was introduced into the sealed reactors to decompose the generated H_2O_2 . The gas was extracted and

determined by GC-MS. The result showed that $^{16}\text{O}_2$ and $^{18}\text{O}_2$ were generally equivalent (**Figure S44**), indicating that two-electron water oxidation was conducted during H_2O_2 production in COF-N32 system. In addition, oxygen evolution reaction was also performed. By using NaBrO_3 as an electron scavenger in Ar atmosphere, negligible amount of O_2 was generated by COF-N32 after 3 h of visible light irradiation (**Figure S43**). The observation further confirms that the water oxidation in COF-N32 system is a two-electron pathway, rather than four-electron pathway.

The corresponding explanations have been added into the revised manuscript (Please see **line 242-247**).

“... while the negligible O_2 generation in oxygen evolution experiment excluded four-electron water oxidation (**Figure S43**). The isotopic experiment was further conducted by using H_2^{18}O and $^{16}\text{O}_2$ as precursors in a sealed reactor. The result shows that the amount of $^{16}\text{O}_2$ and $^{18}\text{O}_2$ generated from the decomposition of H_2O_2 is generally equivalent (**Figure S44**), which further confirms the presence of two-electron water oxidation as well as the charge conservation with oxygen reduction during the H_2O_2 production process.”

Figure S43. The oxygen evolution by COF-N32 in the presence of NaBrO_3 under Ar atmosphere. Conditions: $\lambda > 420$ nm (298K; xenon lamp, light intensity: $100 \text{ mW}\cdot\text{cm}^{-2}$), ultrapure water (50 mL).

Figure S44. Isotopic experiment by using H_2^{18}O as water source during H_2O_2 photosynthesis

Comment 4: Taking into account that the photo-generated holes could oxidize water to form hydroxyl radicals, could the formation of hydroxyl radicals affect the H_2O_2 production reaction?

Response 4: Thank you for your valuable comment. Photo-generated holes in some photocatalysts (e.g. TiO_2) with low valence band potentials could oxidize water to free hydroxyl radicals with $E(\cdot\text{OH}/\text{H}_2\text{O}) = 2.72 \text{ V}$ vs NHE. However, in this study, the valence band potential of COF-N32 is determined to be 1.88 V vs NHE, which is much smaller than $E(\cdot\text{OH}/\text{H}_2\text{O})$. Thus, the oxidation of H_2O to hydroxyl radicals is not available in thermodynamics. Consistently, no obvious signal for $\cdot\text{OH}$ can be detected by using ESR analysis (**Figure S37**). Moreover, the introduction of TBA (the scavenger of diffusing $\cdot\text{OH}$) does not obviously affect the H_2O_2 photosynthesis (**Figure 5a**), which also confirms the insignificant role of diffusing $\cdot\text{OH}$ radicals. The corresponding explanations have been added into the revised manuscript (Please see **lines 227-229**).
 “Meanwhile, the introduction of tertiary butanol (TBA) has negligible effect on H_2O_2 yield by COF-N32 ($p > 0.1$), indicating that diffusing $\cdot\text{OH}$ does not have contribution to the photocatalytic process of H_2O_2 production.”

Figure S38. ESR spectra for DMPO·OH in water before and after visible light irradiation. The typical quartet peaks for DMPO·OH can hardly be observed in the ESR spectra of COF-N32 before and after light illumination, suggesting the absence of diffusing ·OH.

Figure 5. Photocatalytic mechanisms for H₂O₂ production process. (a) Quenching experiments for H₂O₂ photosynthesis. (b) Time-course *in-situ* DRIFT spectra of O₂ on COF-N32 and COF-N31 under visible light irradiation with O₂. (c) PDOS of COF-N31, COF-N32 and COF-N33. The dashed lines stand for the Fermi level. (d) Calculated energy profile for (d) oxidation of water into H₂O₂ and (e) reduction of oxygen into H₂O₂ on COF-N31, COF-N32 and COF-N33 at U = 0 V vs. SHE at pH = 7.

Comment 5: In order to make the manuscript more general, the authors need to include these recent references. ACS Catal. 2022, 12, 12954-12963; Chemical Engineering Journal 454 (2023) 139929.

Response 5: Thank you for your valuable suggestion. We have included the references in the revised manuscript.

Response to Reviewer 3 :

Overall comments: The authors synthesized a class of D-A COFs with different intramolecular polarity by introducing suitable amounts of phenyl groups as electron donors for excitonic regulation to boost the direct photocatalytic H₂O₂ production from water, air, and sunlight without using sacrificial agent. The optimal COF-N32 can facilitate excitons formation and dissociation, leading to the record-high H₂O₂ yield (702 $\mu\text{mol g}^{-1} \text{h}^{-1}$) with SCC of 0.31%. In addition, COF-N32 can also be verified the potential application feasibility with high photocatalytic stability. H₂O₂ photosynthesis is hot topic, and the photocatalytic performance reported in this manuscript is impressive. As a result, this reviewer would like to recommend the acceptance of this manuscript for publication in Nat. Commun. after clarifying the following issues.

Response: Thanks so much for your great efforts in reviewing our revised manuscript. We sincerely appreciate your valuable comments and suggestions. We have revised the manuscript according to your valuable suggestions as well as those from other Reviewers. We believe that the revised manuscript has been substantially strengthened. We are looking forward to your continuous support for our revised manuscript.

Comment 1: In the paragraph describing “Photocatalytic H₂O₂ production by COFs”, authors try to convince the readers of the best photocatalytic performance of COF-N32. They said those “H₂O₂ yield by COF-N32 reaches 702 $\mu\text{mol g}^{-1} \text{h}^{-1}$ ” and “which is much higher than those of recently reported photocatalysts in pure water under same measurement conditions (Figures 3b and S22, Table S3).” However, the given H₂O₂ yield was 3168 $\mu\text{mol g}^{-1} \text{h}^{-1}$ in Figure 3b, which is taken form that in Table S3 and doesn't appear even once in the text. The amount of COF-N32 seems to lead to different yields, however there is no corresponding description in the text. Explanation should be provided for the graphic and text discrepancies.

Response: Thank you for your careful reading and value comment. The Reviewer is correct that lowering the dosage of photocatalysts would improve the mass activity. The H₂O₂ yields by COF-N32 is higher than other reported photocatalysts under the same dosage. Following your excellent suggestion, we have added the corresponding text in the main manuscript (Please see **lines 170-172** and **Figure S24**). “Moreover, COF-N32 can yield over 3.17 mmol $\text{g}^{-1} \text{h}^{-1}$ with the addition of 1 mg COF-N32 in 50 mL ultrapure water (**Figure S24**)...”

Figure S24. H₂O₂ yield by COF-N32 with different dosage in 50 mL O₂-saturated ultrapure water. Conditions: $\lambda > 420 \text{ nm}$ (298K; xenon lamp, light intensity: $100 \text{ mW} \cdot \text{cm}^{-2}$), ultrapure water (50 mL). COF-N32 can yield

over $3168 \mu\text{mol g}^{-1} \text{h}^{-1}$ and $1612 \mu\text{mol g}^{-1} \text{h}^{-1}$ with the addition of 1 and 5 mg COF-N32 in 50 mL ultrapure water, respectively, which are much higher than those of recently reported photocatalysts in pure water under the same measurement conditions (**Table S3**).

Comment 2: The morphology of COFs is suggested to be further characterized by Transmission electron microscopy.

Response 2: Thank you for your valuable comment. Following your excellent suggestion, we have added the TEM images in the revised Supporting Information (Please see **Figure S4**). The corresponding texts have been added into the revised manuscript (Please see **Lines 72-74**).

“Scanning electron microscopy (SEM) and transmission electron microscopy (TEM) images show that the three COFs are tiny granular particles with diameter of 2~3 μm , which are assembled by numerous nanorods (**Figures S3 and S4**).”

Figure S4. TEM images of (a, b) COF-N31, (c, d) COF-N32 and (e, f) COF-N33. The diffraction fringes can not be observed in COF-N31 and COF-N32 due to the relatively low crystallinity of COFs compared with inorganic semiconductors (*J. Am. Chem. Soc.*, 2017, 139, 13083–13091; *J. Am. Chem. Soc.*, 2019, 141, 6152–6156; *Chem. Commun.*, 2019, 55, 167-170). In COF-N33 with high crystallinity, the diffraction fringes of (100) and (002) planes can be found in **Figure S4e** and **S4f**, respectively.

Comment 3: Rotating ring-disk electrode measurements are suggested to be carried out to explore the number of electrons transferred during the catalytic reaction.

Response 3: Thank you for your valuable comment. Following the Reviewer’s excellent suggestion, rotating ring-disk electrode analysis is used to determine the number of electrons transfer from COF-N32 to O_2 . The result shows that the n was 2.17 for COF-N32, suggesting that the oxygen reduction reaction was apparent 2-electron reaction.

The corresponding texts have added in the revised manuscript (Please see **lines 214-217**).

“Rotating ring-disk electrode (RRDE) analysis shows that the number of electrons transferred from COF-N32

to O_2 is estimated to be 2.17 (Figure S37), indicating that O_2 is generally reduced to generate H_2O_2 via the apparent 2-electron reaction. The intermediates during oxygen reduction were further investigated by the trapping experiments.”

Figure S37. (a) RRDE curves over COF-N32-coated electrodes measured at 1600 rpm in O_2 -saturated electrolyte using the ring current (top) and the disk current (bottom). (b) The average number of the transferred electrons (n) at different potentials calculated from RRDE data.

Electrochemical measurement

“Rotating ring-disk electrode (RRDE) analysis was conducted in a three-electrode cell by using Pt foil as a counter electrode and using Ag/AgCl as a reference electrode, respectively. The RRDE was consist of a glassy carbon disk and Pt ring. Before the experiment, COF-N32 was dropped onto the glassy carbon disk and then dried.”

Comment 4: The standardization of various abbreviations and spelling should be carefully checked. For example, “COF-32” in lines 28 and 242 would be corrected to “COF-N32”.

Response 4: Thank you for your pointing out our omission. Following your valuable comment, we have corrected the mistakes in the revised manuscript.

Response to Reviewer 4 :

Overall comments: In this paper, Tong et al. report a strategy to regulate the intramolecular polarity of COFs to enhance the generation and separation of photoinduced excitons, thus optimizing the performance of photocatalytic H₂O₂ generation by these COFs. They claimed that a COF with a moderate intramolecular polarity that contains a triazine core and two peripheral benzenes exhibited the greatest performance for H₂O₂ production. The generalizability of this COF photocatalyst was verified by varying the water source, assembling it into practical devices and reusing for several times, which gives satisfactory performance in all the cases. The authors investigated the mechanism of H₂O₂ production by conducting quenching experiments, in-situ ESR analysis, in-situ FTIR analysis and DFT calculations, elucidating the simultaneous oxygen reduction and water oxidation pathways and revealing the triazine nitrogen and carbon as the active sites for water oxidation and O₂ reduction, respectively. In summary, a comprehensive structure-property-function relationship was established in this paper, which provides a clear clue for the design of COFs *via* regulating their intramolecular polarity to boost photocatalytic performance. The whole manuscript is arranged logically and well-written. The figures and illustrations are appropriate for interpretation. The conclusion was well supported by the experimental data and analysis. Thus, I think this manuscript meets the criteria of Nature Communications and I recommend the publication of this manuscript after addressing the following concerns.

Response: Thanks so much for your great efforts in reviewing our revised manuscript. We sincerely appreciate your valuable comments and suggestions. We have revised the manuscript according to your valuable suggestions as well as those from other Reviewers. We believe that the revised manuscript has been substantially strengthened. We are looking forward to your continuous support for our revised manuscript.

Comment 1: The author claimed the record-high H₂O₂ yield (702 μmol g⁻¹ h⁻¹) for a COF photocatalyst in the absence of a sacrificial reagent. From the viewpoint of charge conservation, theoretically, the electrons generated on COF which reduce oxygen should come from the oxidation of water. In order to exclude the self-oxidation of COF itself, H₂O¹⁸ is suggested to use to verify the water oxidation process. The amount of 2 * ¹⁸O₂ and H₂¹⁸O₂ in the system should be equal to the H₂¹⁶O₂ to meet the requirement of charge conservation. Otherwise, self-oxidation of COF cannot be excluded.

Response 1: Thank you for your valuable comment. Following your excellent suggestion, ¹⁸O isotopic experiment was performed to confirm the charge conservation between water oxidation and oxygen reduction. Typically, the H₂O₂ photosynthesis by COF-N32 was conducted in H₂¹⁸O under ¹⁶O₂ atmosphere. After the visible irradiation, N₂ was purged into the reaction system to remove excessive ¹⁶O₂ and air in the sealed reactor, followed by the injection of catalase to decompose the produced H₂O₂ into O₂. The gas was extracted and injected into GC-MS for analysis. The result showed that ¹⁶O₂ and ¹⁸O₂ were generally equivalent (**Figure S44**). The observation indicates that the amount of H₂¹⁶O₂ is equal to the amount of H₂¹⁸O₂, which confirms the charge conservation between water oxidation and oxygen reduction.

The corresponding explanations have been added into the revised manuscript (Please see **lines 243-248** in main manuscript and **lines 163-168** in SI).

“...while the negligible O₂ generation in oxygen evolution experiment excluded four-electron water oxidation (**Figure S43**). The isotopic experiment was further conducted by using H₂¹⁸O and ¹⁶O₂ as precursors in a sealed reactor. The result shows that the amount of ¹⁶O₂ and ¹⁸O₂ generated from the decomposition of H₂O₂ is generally equivalent (**Figure S44**), which further confirms the presence of two-electron water oxidation as well as the charge conservation with oxygen reduction during the H₂O₂ production process.”

Figure S44. Isotopic experiment by using H_2^{18}O as water source during H_2O_2 photosynthesis

^{18}O isotopic experiment

^{18}O isotopic experiment on COF-N32 was performed by using H_2^{18}O with saturated $^{16}\text{O}_2$ in a sealed reactor, which was irradiated by Xenon lamp (>420 nm). After H_2O_2 photosynthesis, the reaction suspension was purged by N_2 to remove $^{16}\text{O}_2$ in the reactor. Subsequently, the photo-generated H_2O_2 was decomposed into O_2 by adding catalase. The evolved O_2 gas was analyzed by a gas chromatography-mass spectrometry (GC-MS, Agilent 7890B-5977B)."

Comment 2: "The signal intensities of three COFs follow the order of COF-N31 > COF-N32 > COF-N33 (Figure S5), which might result from the difference in intramolecular polarity among three COFs". As we know, the ESR signal comes from unpaired electrons in the materials. How do the authors draw a conclusion that different ESR signal intensities result from the different polarities in COFs? Is there any relationship between unpaired electrons and intramolecular polarity?"

Response 2: Thank you for pointing out our omission. The solid-state ESR signal should be attributed to partial charge separation and the production of unpaired electrons, which was driven by the electron push-pull effect of donor-acceptor structure (Liu et al., *ACS Catal.*, 2022, 12, 9494-9502; Yan et al., *Proc. Natl. Acad. Sci. U. S. A.*, 2022, 119, e2202913119). To avoid confusion, the corresponding text has been rewritten in the revised manuscript (Please see **lines 88-89**).

"...indicating that different amounts of unpaired electrons exist in three D-A COFs under dark condition³⁰."

Comment 3: I doubt the relationship between water contact angles and polarity. Because measurement of water contact angles was in macroscale and the polarity was usually described in molecular scale. Also, the measurement of water contact angles will likely be influenced by many factors such as the preparation of samples, material morphology, temperatures, humidity, and so on. Alternatively, water adsorption analysis is suggested to study the polarity because the adsorption usually takes place in micro to mesoscale, which might be more convincing.

Response 3: Thank you for your valuable comment. Following the Reviewer's excellent suggestion, we have also conducted the water adsorption analysis. The result shows that the unit water adsorption capacity also follows the order of COF-N31 > COF-N32 > COF-N33, which is consistent with the order of intramolecular

polarity. The corresponding texts have been added into the revised manuscript (**Lines 106-108**).

“Meanwhile, the unit water adsorption capacity is also consistent with the order of COF-N31 > COF-N32 > COF-N33 (**Figure S10**).”

Figure S10. The water adsorption isotherms of COF-N31, COF-N32 and COF-N33, which is normalized by BET surface area.

Comment 4: “Among three COFs, COF-N32 exhibits smaller average lifetime (86 ps) relative to COF-N31...” what kind of species do the lifetimes represent? photoexcited electrons? Holes? Or excitons?

Response 4: Thank you for your valuable comment. The positive absorbance changes are attributed to the photoexcited electrons of three COFs. Therefore, the lifetimes are corresponded to the photoexcited electrons. The corresponding texts have been rewritten in the revised manuscript (Please see **lines 146-147**). “Among three COFs, photoexcited electrons in COF-N32 exhibits smaller average lifetime (86 ps) relative to those in COF-N31 (116 ps) and COF-N33 (225 ps) (**Figure 2c**)...”

Comment 5: The author performed the isopropanol (IPA) quenching experiment to trap the diffusing ·OH. However, isopropanol might act as a sacrificial agent which directly extracts holes. So is this appropriate to use IPA as the trapping agent of diffusing ·OH?

Response 5: Thank you for your valuable comment. We agree with the Reviewer that high concentration of IPA would react with holes and might not be appropriate to quench diffusing ·OH. According to previous studies, tertiary butanol (TBA) can be the scavenger of diffusing ·OH with the high kinetic constants of $3.8\text{--}7.6 \times 10^8 \text{ M}^{-1} \text{ s}^{-1}$. Thus, we also perform the quenching experiment by using TBA. The result shows that no obvious influence of TBA on the H₂O₂ photosynthesis by COF-N32 ($p > 0.1$), indicating that diffusing ·OH plays negligible role in the photocatalytic H₂O₂ production process by COF-N32. The corresponding texts have been rewritten in the revised manuscript (Please see **lines 227-229**). “Meanwhile, the introduction of tertiary butanol (TBA) has negligible effect on H₂O₂ yield by COF-N32 ($p > 0.1$), indicating that diffusing ·OH does not contribute to the photocatalytic process of H₂O₂ production.”

Figure 5. Photocatalytic mechanisms for H₂O₂ production process. (a) Quenching experiments for H₂O₂ photosynthesis. (b) Time-course *in-situ* DRIFT spectra of O₂ on COF-N32 and COF-N31 under visible light irradiation with O₂. (c) PDOS of COF-N31, COF-N32 and COF-N33. The dashed lines stand for the Fermi level. (d) Calculated energy profile for (d) oxidation of water into H₂O₂ and (e) reduction of oxygen into H₂O₂ on COF-N31, COF-N32 and COF-N33 at U= 0 V vs. SHE at pH = 7.

Figure S36. Effects of scavengers on the H₂O₂ photosynthesis kinetics by COF-N32 under visible light irradiation.

Comment 6: In Figure 2a, two transitions can be observed for COF-N33, which is different from COF-N31 and N32. How to explain this as the three COFs are anal

Response 6: Thank you for your valuable comment. All three COFs exhibit n-π* transition (particularly in triazine N atoms), while COF-N33 with more benzene units also contains more obvious signals of π-π* transition. Similar observation has been reported in the previous study about anthracene-containing COFs (*Angew. Chem. Int. Ed.*, 2015, 54, 8704-8707). The corresponding texts have been added into the revised main manuscript (**Lines 129-130**).

“Tauc plot based on UV-vis diffused reflectance spectra (DRS) reveals that all three COFs exhibit n-π* transition in N atoms³⁹, while COF-N33 with more benzene units also contains more obvious signals of π-π* transition⁴⁰.”

Response to Reviewer 5 :

Overall comments: In this work, the authors have synthesized and characterized a set of three covalent organic frameworks (COFs) that show a cost-effective approach to producing H₂O₂ from the air, water, and sunlight. In addition, they present a strategy for designing metal-free COFs with optimal properties to produce H₂O₂. The work is interesting, but there is still some additional work that needs to be done before its suitable for publication. As a result, I cannot recommend it for publication at the moment. My detailed comments are provided below:

Response: Thanks so much for your great efforts in reviewing our revised manuscript. We sincerely appreciate your valuable comments and suggestions. We have revised the manuscript according to your valuable suggestions as well as those from other Reviewers. We believe that the revised manuscript has been substantially strengthened. We are looking forward to your continuous support for our revised manuscript.

Comment 1: The authors need to be explicit in the text about how they quantified the H₂O₂ that was produced. It was sort of frustrating to dig for this vital information.

Response 1: Thank you for your pointing out our omission. We have added the corresponding texts in the revised Supporting Information (Please see **lines 70-76** in SI).

The measurement of H₂O₂

“The production of H₂O₂ was measured by an iodometry method according to the literature (*Energy & Environmental Science*, 2018, 11, 2581-2589). Specifically, 1 mL of samples was added to the mixture of 1 mL of 0.4 M KI and 1 mL 0.1 M potassium hydrogen phthalate (C₈H₅KO₄), which was kept for 1 h. Under acidic conditions, H₂O₂ can react with I⁻ to generate triiodide anions (I₃⁻), which exhibited absorption at 350 nm. Thus, the absorbance at 350 nm by using UV-vis spectroscopy can measure the amount of I₃⁻, which can further determine the amount of H₂O₂ produced in each sample.”

Comment 2: The authors should check the production of hydrogen and oxygen since the band gap suggests this material should be capable of oxidizing water to O₂ and H₂.

Response 2: Thank you for your valuable comment. Following the Reviewer’s valuable comment, we have tested the production of H₂ during the H₂O₂ photosynthesis in a sealed reactor. However, no production of H₂ could be observed (**Figure R3**), indicating that H⁺ preferred to react with O₂/·O₂⁻ in COF-N32 system. In addition, we used NaBrO₃ as the electron scavenger to investigate the oxygen evolution reaction on COF-N32 (**Figure S43**). Yet, O₂ could be hardly produced via the four-electron pathway. Instead, two-electron water oxidation was conducted on COF-N32 to generate H₂O₂.

Figure R3. The hydrogen evolution by COF-N32. Conditions: $\lambda > 420$ nm (298K; xenon lamp, light

intensity:100 mW·cm⁻²), ultrapure water (50 mL).

Figure S43. The oxygen evolution by COF-N32 in the presence of NaBrO₃ under Ar atmosphere. Conditions: $\lambda > 420$ nm (298K; xenon lamp, light intensity:100 mW·cm⁻²), ultrapure water (50 mL).

Comment 3: The authors should refrain from using SEM to qualify the porous nature of these materials. SEM would describe the morphology of the material. If the authors are adamant about using an imaging technique to report the pore size of the COFs, suggest they obtain high-resolution TEM. Surprisingly, the author uses an imaging technique to qualify the pore size distribution of the material when the acceptable standard is to use BET analysis. Since the authors have already conducted BET measurements for these COFs to quantify the surface area of these COFs, they should report the pore-size distribution from the BET analysis.

Response 3: Thank you for your valuable comment. We agree the Reviewer that SEM could only show morphology, instead of porous structure. Following the Reviewer's excellent suggestion, we have added high-resolution TEM in **Figure S4**. The pore size distribution is also provided in **Figure S48**, which shows the micropore structure of three COFs. The corresponding text has been added in the revised manuscript (Please see **line 71-73** and **Figure S48**).

“Scanning electron microscopy (SEM) and transmission electron microscopy (TEM) images show that the three COFs are tiny granular particles with diameter of 2~3 μm, which are assembled by numerous nanorods (**Figures S3 and S4**).”

Figure S4. TEM images of (a, b) COF-N31, (c, d) COF-N32 and (e, f) COF-N33. The diffraction fringes can not be observed in COF-N31 and COF-N32 due to the relatively low crystallinity of COFs compared with inorganic semiconductors (*J. Am. Chem. Soc.*, 2017, 139, 13083–13091; *J. Am. Chem. Soc.*, 2019, 141, 6152–6156; *Chem. Commun.*, 2019, 55, 167–170). In COF-N33 with high crystallinity, the diffraction fringes of (100) and (002) planes can be found in **Figure S4e** and **S4f**, respectively.

Figure S48. N₂ adsorption-desorption isotherms of (a) COF-N31, (b) COF-N32 and (c) COF-N33 and the corresponding pore size distribution (insets). (d) O₂-TPD curves of three COFs. The BET surface area of COF-N31, COF-N32 and COF-N33 is determined to be 81 m²/g, 823 m²/g and 677 m²/g, respectively. The three COFs shows narrow pore size of 1-2 nm, indicating their micropore structure. The intensities of O₂-TPD signals follow the order of COF-N31 > COF-N32 > COF-N33. It should be noted that the COFs are thermally stable at the tested temperature (< 200°C, **Figure S47**), suggesting the signals in O₂-TPD are not resulted from the decomposition of COFs. Therefore, the O₂ adsorption is mainly relevant to the structure of COFs instead of surface area. Besides, COF-N31 with more O₂ adsorption exhibits relatively low production capability of 1,4-endoperoxide species and adsorbed ·O₂⁻ compared with that of COF-N32, further implying that the electron transfer efficiency in oxygen reduction is more important than surficial properties during H₂O₂ production process.

Comment 4: The authors should obtain a better PXRD profile. I suggest using a Voltage of 30 and a Current of 20 and running the experiment for at least 40 minutes. Running PXRD with high power induces fluoresces in the material, resulting in poor peaks.

Response 4: Thank you for your valuable comment. Following the Reviewer's excellent suggestion, we have performed XRD analysis using a voltage of 30 kV and a current of 20 mA. The quality of PXRD pattern is improved (**Figure S2**).

Figure S2. Experimental and simulated X-ray diffraction (XRD) patterns of (a) COF-N31-DMSO, (b) COF-N31, (c) COF-N32 and (d) COF-N33. Pawley refinements show the good agreement between experimental and simulated eclipsed XRD patterns ($R_{wp} = 2.67\%$, $R_p = 2.10\%$ for COF-N31-DMSO, $R_{wp} = 3.27\%$, $R_p = 2.47\%$ for COF-N32, $R_{wp} = 3.49\%$, $R_p = 2.50\%$ for COF-N33). Since its crystallinity was not very high, the XRD pattern of COF-N31 does not match well with the simulation. Similar observations have been reported in previous studies (*Chem. Commun.*, 2019,55, 167-170, *Chem. Commun.*, 2017,53, 9636-9639). Preliminary experiments show that COF-N31-DMSO with high crystallinity yet exhibits low photo-stability but similar excitonic behavior with COF-N31 (data not shown). Therefore, COF-N31 with the same chemical structure as COF-N31-DMSO is thus used in the following experiments.

Comment 5: The authors should obtain simulated PXRDs of the COFs to determine the interlayer stacking of these materials. This information is crucial because the authors claim that these are novel COFs. Additionally, determining the interlayer stacking will also help in explaining their results.

Response 5: Thank you for your valuable comment. Following the Reviewer's excellent suggestion, the simulation and Pawley refinement have been done. Pawley refinements show the good agreement between experimental and simulated eclipsed XRD patterns ($R_{wp} = 2.67\%$, $R_p = 2.10\%$ for COF-N31-DMSO, $R_{wp} = 3.27\%$, $R_p = 2.47\%$ for COF-N32, $R_{wp} = 3.49\%$, $R_p = 2.50\%$ for COF-N33). The revised **Figure S2** can be found in the Response to your Comment 4.

REVIEWER COMMENTS

Reviewer #1 (Remarks to the Author):

The authors did some modifications to the manuscript according to my suggestions. A new COFs named COF-N31-DMSO was prepared to solve the structure analysis problem. However, as the author claimed, COF-N31-DMSO could not be used as the catalyst due to its poor stability. The catalysis study was still done by COF-N31 which has a totally different PXRD pattern from COF-N31-DMSO. This is a serious problem. The structure analysis of COF-N31 is basically still missing. The simulation failed to give the structure information of COF-N31. However, the COF-N31 is the one that is used to compare the catalysis performance. This problem must be addressed before publication. Either clarify the structure or give a new description of COF-N31 avoiding using the name "COFs".

Reviewer #2 (Remarks to the Author):

I recommend this manuscript be published in Nature Communications with this version.

Reviewer #3 (Remarks to the Author):

Since all the issues concerned have been clarified in the present version of manuscript, I suggest acceptance of this paper for publication in Nature Communications.

Reviewer #4 (Remarks to the Author):

In this revised version, the authors have carefully made corrections and responses according to the Reviewers' comments, therefore, I recommend accepting this work for publication in this Journal at its current state.

Reviewer #5 (Remarks to the Author):

My comments have been addressed and I'm satisfied with the revised manuscript.

Reviewer 1 :

Overall comments: The authors did some modifications to the manuscript according to my suggestions. A new COFs named COF-N31-DMSO was prepared to solve the structure analysis problem. However, as the author claimed, COF-N31-DMSO could not be used as the catalyst due to its poor stability. The catalysis study was still done by COF-N31 which has a totally different PXRD pattern from COF-N31-DMSO. This is a serious problem. The structure analysis of COF-N31 is basically still missing. The simulation failed to give the structure information of COF-N31. However, the COF-N31 is the one that is used to compare the catalysis performance. This problem must be addressed before publication. Either clarify the structure or give a new description of COF-N31 avoiding using the name “COFs”

Response: We sincerely appreciate your valuable comments and suggestions. We agree with the Reviewer that the original COF-N31 fabricated in mesitylene/dioxane could not be considered as COFs since it has different PXRD pattern with structural simulations of COFs due to its weak PXRD peaks. The catalyst fabricated in DMSO/DMAc system based on the literature suggested by the Reviewer (*J. Am. Chem. Soc.*, 2019, 141, 6152–6156) actually is COF-N31 since its PXRD pattern matches well with structural simulations. **The newly fabricated COF-N31 in DMSO/DMAc system thus was well characterized using different techniques including XPS (Figure S1), XRD (Figure S2), ¹³C NMR spectra (Figure 1b), FTIR (Figure 1c), SEM (Figure S3), TEM (Figure S4), ESR (Figure S6), UV-vis DRS (Figures 2a and S8), water contact angles (Figure S9), water adsorption (Figure S10), EIS (Figure S13), PL (Figures S15 and S17), transient adsorption spectra (Figures 2c and S16) as well as temperature-dependent PL (Figures 2b and S14), and so on. The newly fabricated COF-N31 was used for the catalysis study including photosynthesis of H₂O₂ (Figures 3a and S22), reused cycles experiments (Figure S26) and isotopic experiment by using H₂¹⁸O (Figure S27).**

It is worth pointing out that updating the corresponding results with those from newly prepared COF-N31 did not affect our conclusion about the relationship between intramolecular polarity and exciton behaviors.

We have replaced the description and the results about COF-N31 with those of the newly fabricated material in the 2nd revised manuscript.:

“... which are assembled by numerous nanorods (Figures S3 and S4).” (Line 74)

“Accordingly, stable radicals with strong signal intensities are observed in the solid-state electron spin resonance (ESR) spectra ... (Figure S6)” (Line 87)

“Obvious shift of emission peaks (~48 nm) ... (Figure S8)” (Line 97)

“... COF-N31 (121°) ... (Figure S9)” (Line 106)

“Meanwhile, the unit water adsorption capacity is also consistent with the order of COF-N31 > COF-N32 > COF-N33 (Figure S10).” (Lines 107-108)

“The band gaps of COF-N31 (2.72 eV) ... (Figure 2a)” (Line 131)

“The results of electrical impedance spectra (EIS) indicate ... (Figure S13)” (Lines 135-136)

“Both the exciton binding energy (E_b) determined by temperature-dependent photoluminescence (PL) spectra and the emission peaks intensities in steady-state PL spectra of three COFs follow the order of COF-N33 > COF-N32 > COF-N31 (Figures 2b, S14 and S15)” (Lines 139-141)

“...can be attributed to the photoinduced absorption of photoexcited electrons in the conduction band (CB) of three COFs (Figure S16)” (Lines 145-146)

“... relative to those in COF-N31 (46 ps) and COF-N33 (52 ps) (Figure 2c) ...” (Line 147)

“...relative to 0.75 ns for COF-N31 ... (Figure S17)” (Line 148)

“... COF-N32 can generate more photo-induced free charges (via the excitons formation and dissociation) than COF-N31 (Figure S18a) and COF-N33 (Figure S18b).” (Lines 151-152)

“... -1.09 V vs. NHE for COF-N31 ... (Figure S20 and S21)” (Line 159)

“Specifically, after 12 h of visible light irradiation, H₂O₂ yield by COF-N32 reaches 7092 $\mu\text{mol g}^{-1}$ (605 $\mu\text{mol g}^{-1} \text{h}^{-1}$), which is greatly higher than that by COF-N31 (4316 $\mu\text{mol g}^{-1}$, 434 $\mu\text{mol g}^{-1} \text{h}^{-1}$) and COF-N33 (1736 $\mu\text{mol g}^{-1}$, 155 $\mu\text{mol g}^{-1} \text{h}^{-1}$) (Figures 3a and S22)” (Lines 169-172)

Moreover, the H₂O₂ production by both COF-N31 and COF-N33 during reused cycles were also investigated. The crystalline structure of both COF-N31 and COF-N33 after use was also characterized (**Figure S26**). We found that in pure water, COF-N31 with strong intramolecular polarity has low photo-stability of COF-N31 during the reaction duration (**Figure S26-28**), while COF-N32 and COF-N33 with relatively weak intramolecular polarity especially COF-N32 contain excellent photo-stability under visible light irradiation and can be consecutively reused for the photo-generation of H₂O₂. The corresponding texts have been incorporated in the 2nd revised manuscript:

*“The stable H₂O₂ yield during the 5 reused cycles and no obvious change of crystalline structure after use are also achieved for COF-N33 (**Figures S26a and S26b**). In contrast, the H₂O₂ yield by COF-N31 dramatically decreases with increasing reused cycles (**Figure S26c**), indicating the relatively low photo-stability of COF-N31 during the reaction duration. The decreased crystallinity of COF-N31 after photocatalytic reaction (**Figure S26d**) confirms its low photo-stability during the reaction process...The results clearly show that in pure water, COF-N31 with strong intramolecular polarity has low photo-stability of COF-N31 during the reaction duration, whereas COF-N32 and COF-N33 with relatively weak intramolecular polarity especially COF-N32 owns excellent photo-stability under visible light irradiation and can be consecutively reused for the photo-generation of H₂O₂.” (Lines 184-189, 204-208)*

“COF-N31 was fabricated by solvothermal methods according to the literature²². Specifically, 0.3 mmol of 1,3,5-triformylphloroglucinol (Tp, 63 mg) and 0.3 mmol of melamine (38 mg) were added into a reactor, followed by the addition of 2 mL of dimethyl sulfoxide, 1 mL of N,N-dimethylacetamide (DMAc) and 0.3 mL of 6 M acetic acid. After ultrasonication and degassed by three consecutive freeze-pump-thaw cycles, the reactor was sealed under vacuum condition, which was then heated at 120 °C for 72 h. The collected product was firstly rinsed by DMAc, which was then solvent exchanged with DMAc, pure water and washed with acetone for three times. The final product was dried at 120 °C under vacuum.” (Lines 321-327)

The figures with updated results of the newly prepared COF-N31 in the 2nd revised manuscript are listed as follows:

Figure 1. Chemical structure. (a) Schematic illustration of synthesis process of COF-N31, COF-N32 and COF-N33. (b) ^{13}C NMR spectra and (c) Fourier transformation infrared spectroscopy (FTIR) of COF-N31, COF-N32 and COF-N33. (d) The schematic illustration of the octupolar structure in COF-N31, COF-N32 and COF-N33. (e) DFT calculation on charge distribution of COF-N32 structure.

Figure 2. Polarity property. (a) Tauc plot of three COFs; (b) Temperature-dependent PL spectra of three COFs excited at 365 nm. (c) photoinduced absorption decay dynamics of three COFs with the excitation of 360 nm pump pulse ($P = 4.3 \text{ mJ cm}^{-2}$ per pulse); (d) The signal intensity of TEMPO for charge detection in COF-N32 under light irradiation.

Figure 3. Photocatalytic performance. (a) Photosynthesis of H_2O_2 by COF-N31, COF-N32 and COF-N33. Experimental conditions: $\lambda > 420 \text{ nm}$ (298K; xenon lamp, light intensity: $100 \text{ mW} \cdot \text{cm}^{-2}$), ultrapure water (50 mL), photocatalyst (25 mg). (b) The comparison of H_2O_2 photosynthesis rate by COF-N32 with other reported photocatalysts without sacrificial agent under the similar measurement conditions. (c) The apparent quantum yield (AQY) of COF-N32 as a function of wavelength (purple light at 400 nm, blue light at 459 nm, green light at 519 nm, yellow light at 580 nm, red light at 625 nm). (d) The reusability of COF-N32 for H_2O_2 photosynthesis. Experimental conditions: $\lambda > 420 \text{ nm}$ (298K; xenon lamp, light intensity: $100 \text{ mW} \cdot \text{cm}^{-2}$), ultrapure water (50 mL), photocatalyst (25 mg).

Figure 5. Photocatalytic mechanisms for H₂O₂ production process. (a) Quenching experiments for H₂O₂ photosynthesis. Conditions: $\lambda > 420$ nm (298K; xenon lamp, light intensity: $100 \text{ mW} \cdot \text{cm}^{-2}$), volume (50 mL), photocatalyst (25 mg), $[\text{p-BQ}]_0 = 3 \text{ mM}$, $[\text{TBA}]_0 = 3 \text{ mM}$. (b) Time-course *in-situ* DRIFT spectra of O₂ on COF-N32 under visible light irradiation with O₂. (c) PDOS of COF-N31, COF-N32 and COF-N33. The dashed lines stand for the Fermi level. (d) Calculated energy profile for (d) oxidation of water into H₂O₂ and (e) reduction of oxygen into H₂O₂ on COF-N31, COF-N32 and COF-N33 at $U = 0 \text{ V vs. SHE}$ at pH = 7.

Figure S1. XPS spectra of three COFs. (a) Survey scan XPS profile, high-resolution XPS spectra of (b) O 1s and (c) N 1s of COFs. The O 1s peaks can be divided into two peaks for absorbed -OH at 531.1 eV and C=O at 533.0 eV, respectively. The peaks at 398.6 eV and 400.0 eV in N 1s spectra of COFs can be indexed to C-NH- and C-N=C, respectively.

Figure S3. SEM images of (a) COF-N31, (b) COF-N32 and (c) COF-N33. SEM images show the spherical structure of three COFs with a diameter of 2~3 μm , which is assembled by numerous nanorods.

Figure S4. TEM images of (a, b) COF-N31, (c, d) COF-N32 and (e, f) COF-N33. The diffraction fringes can not be observed in COF-N31 and COF-N32 due to the relatively low crystallinity of COFs compared with inorganic semiconductors^{S1, S4-5}. In COF-N33 with high crystallinity, the diffraction fringes of (100) and (002) planes can be found in **Figure S4e** and **S4f**, respectively.

Figure S6. Solid-state electron spin resonance spectra of three COFs under dark and ambient conditions.

Figure S8. Solvatochromic behaviors of (a) COF-N31, (b) COF-N32 and (c) COF-N33.

Figure S9. Water contact angles (CA) of water droplet on the pressed pellet of (a) COF-N31, (b) COF-N32 and (c) COF-N33.

Figure S10. The water adsorption isotherms of three COFs, which is normalized by BET surface area.

Figure S13. Electric impedance spectra of three COFs.

Figure S14. Integrated PL intensities as a function of temperature of (a) COF-N31, (b) COF-N32 and (c) COF-N33.

Figure S15. Steady-state PL spectra of three COFs excited at 365 nm.

Figure S16. Transient adsorption spectra of three COFs registered at different probe delays (pump at 360 nm).

Figure S17. PL decay curves of three COFs.

Figure S18. ESR spectra of TEMPO for the detection of photo-generated free charges in (a) COF-N31 and (b) COF-N33.

Figure S20. Valence band XPS spectra of three COFs. The equilibration of the Fermi level of the instrument was conducted at 4.5 eV by using Au metal basis as the reference. The potentials calculated by binding energy in VB-XPS spectra equals to the VB potential (vs. NHE). Thus, the VB potentials of COF-N31, COF-N32 and COF-N33 are determined to be 1.63 V vs. NHE, 1.88 V vs. NHE and 1.92 V vs. NHE, respectively.

Figure S21. Schematic illustration of energy bands of three COFs. According to the VB-XPS results and the band gaps calculated in Tauc plot (Figure 2a), the energy band position of COFs can be determined, which confirms that the oxygen reduction reaction by COFs are thermodynamically feasible. The two-electron water oxidation reaction (WOR) directly to H_2O_2 (1.77 V vs. NHE) is also thermodynamically possible by COF-N32 (1.88 V vs. NHE) and COF-N33 (1.92 V vs. NHE), but not by COF-N31 (1.63 V vs. NHE). Instead, h^+ generated by COF-N31 can attack COF itself in pure water without scavengers.

Figure S22. (a) H_2O_2 photosynthesis kinetics by different photocatalysts under visible light irradiation; (b) H_2O_2 photosynthesis kinetics as a function of intramolecular polarization for three COFs.

Figure S26. The reusability of (a) COF-N33 and (c) COF-N31 for H_2O_2 photosynthesis. XRD patterns of (b) COF-N33 and (d) COF-N31 before and after reaction. Conditions: $\lambda > 420 \text{ nm}$ (298K; xenon lamp, light intensity: $100 \text{ mW} \cdot \text{cm}^{-2}$), ultrapure water (50 mL), photocatalyst (25 mg).

Figure S27. Isotopic experiment by using H_2^{18}O as water source during H_2O_2 photosynthesis by (a) COF-N31, (b) COF-N32 and (c) COF-N33.

Figure S39. *In-situ* ESR spectra of $\text{DMPO} \cdot \text{O}_2^-$ for COF-N31, COF-N32 and COF-N33 under 5 min of visible light irradiation.

Figure S40. Thermogravimetric curves of (a) COF-N31, (b) COF-N32 and (c) COF-N33. Less than 10% weight loss of COFs was observed under 200 °C, which can be attributed to the loss of adsorbed H₂O on the surface of COFs.

Figure S41. N₂ adsorption-desorption isotherms of (a) COF-N31, (b) COF-N32 and (c) COF-N33 and the corresponding pore size distribution (insets). (d) O₂-TPD curves of three COFs. The BET surface area of COF-N31, COF-N32 and COF-N33 is determined to be 75 m²/g, 823 m²/g and 677 m²/g, respectively. The three COFs shows narrow pore size of 1-2 nm, indicating their micropore structure. The intensities of O₂-TPD signals follow the order of COF-N31 > COF-N32 > COF-N33. It should be noted that the COFs are thermally stable at the tested temperature (< 200°C, **Figure S40**), suggesting the signals in O₂-TPD are not resulted from the decomposition of COFs. Therefore, the O₂ adsorption is mainly relevant to the structure of COFs instead of surface area. Besides, COF-N31 with more O₂ adsorption exhibits relatively low production capability of ·O₂⁻ and H₂O₂ compared with that of COF-N32, further implying that the electron transfer efficiency in oxygen reduction is more important than surficial properties during H₂O₂ production process.

Reviewer 2 :

Overall comments: I recommend this manuscript be published in Nature Communications with this version.

Reviewer 3 :

Overall comments: Since all the issues concerned have been clarified in the present version of manuscript, I suggest acceptance of this paper for publication in Nature Communications.

Reviewer 4 :

Overall comments: In this revised version, the authors have carefully made corrections and responses according to the Reviewers' comments, therefore, I recommend accepting this work for publication in this Journal at its current state.

Reviewer 5 :

Overall comments: My comments have been addressed and I'm satisfied with the revised manuscript.

Response to Reviewers 2-5: We appreciate the reviewers' very positive comments. We are looking forward to the continuous support from reviewers for our revised manuscript!

REVIEWERS' COMMENTS

Reviewer #1 (Remarks to the Author):

The authors tried their best to address the concerns. From the viewpoint of chemists, the problem of PXRD for COF-N31 is still open. Considering the catalysis work is still significant, as a materials science-related manuscript, it can be accepted as this.

Reviewer #4 (Remarks to the Author):

This revised version has well-resolved all the concerns from our side, thus I recommend accepting for publication in its current state.

Reviewer 1 :

Overall comments: The authors tried their best to address the concerns. From the viewpoint of chemists, the problem of PXRD for COF-N31 is still open. Considering the catalysis work is still significant, as a materials science-related manuscript, it can be accepted as this.

Response to Reviewer 1: Thank you very much for your positive comment. We are looking forward to your continuous support for our revised manuscript!

Reviewer 4 :

Overall comments: This revised version has well-resolved all the concerns from our side, thus I recommend accepting for publication in its current state.

Response to Reviewer 4: Thank you very much for your recognition and recommendation of our work.